# Laser Imaging Nephelometer for aircraft deployment

Adam T. Ahern[1,2], Frank Erdesz[1,2], Nicholas L. Wagner[1,2]*, Charles A. Brock[1], Ming Lyu[3], Kyra Slovacek[2,4], Richard H. Moore[5], Elizabeth B. Wiggins[5,6], and Daniel M. Murphy[1]

[1]NOAA Chemical Sciences Laboratory, Boulder, CO 80305, USA
[2]Cooperative Institute for Research in Environmental Sciences, University of Colorado, Boulder, CO 80309, USA
[3]Department of Chemistry, University of Alberta, Edmonton, AB T6G 2B4, Canada
[4]Civil, Environmental, and Architectural Engineering, University of Colorado, Boulder, CO 80309, USA
[5]NASA Langley Research Center, Hampton, VA 23666, USA
[6]NASA Postdoctoral Program, Universities Space Research Association, Columbia, MD 21046, USA
*Now at Ball Aerospace, Westminster, CO 80021, USA

*Correspondence to*: Adam T. Ahern (adam.ahern@noaa.gov)

**Abstract.** Validation of remote sensing retrievals of aerosol microphysical and optical properties requires *in situ* measurements of the same properties. We present here an improved imaging nephelometer for measuring the directionality and polarization of light
(i.e. polarimetry) scattered at two wavelengths (405 nm and 660 nm) with high temporal resolution. The instrument was designed for airborne deployment and is capable of ground-based measurements as well. The Laser Imaging Nephelometer (LiNeph) uses two orthogonal detectors with wide-angle lenses and linearly polarized light sources to measure both the phase function, $P_{11}(\theta)$, and degree of linear polarization, $-P_{12}/P_{11}(\theta)$. In this work, we will describe the instrument function and calibration, as well as data acquisition and reduction. The instrument was first deployed aboard the NASA DC-8 during the 2019 FIREX-AQ campaign. Here,
we present field measurements of smoke plumes that show that the LiNeph has sufficient resolution for 0.24 Hz polarimetric measurements at two wavelengths, 405 and 660 nm, at integrated scattering coefficients ranging from $50 - 8,000$ Mm$^{-1}$.

## 1 Introduction

Although greenhouse gases are a dominant climate forcer, tropospheric particles also have large and under-constrained effects on the Earth's radiative budget. To understand these effects, long-term monitoring of particle number, size, and composition with
global coverage is required. Satellite and ground-based remote measurements of light scattered by these particles are the only practical way to achieve this temporal and spatial coverage. The remote measurements with the greatest spatial coverage are those that utilize passive sensors, i.e. those which measure sunlight scattered by the atmosphere and planet surface. It is important to account for observational geometry when retrieving aerosol microphysical and optical properties from scattered light measurements.

Given that existing remote sensors can typically only measure at a few discrete wavelengths and scattering angles, there are many theoretical combinations of particle populations that could explain the observed scattered light. For example, if the sensor can only detect scattered light at one angle and one wavelength, scattered light could be explained by many small particles but also by a few large particles. Additional information like the amount of light scattered at different wavelengths, the polarization state, and scattering intensity at different angles can reduce the number of aerosol populations that can explain the observations, but the
system will still remain underdetermined (Dubovik and King, 2000). Thus, it is often useful to make simplifying assumptions about the particle populations based on prior environmental observations, and then derive and refine important and useful quantities, such as aerosol optical depth and aerosol microphysical properties (Dubovik et al., 2002). For spherical aerosols of known size and composition, Mie theory provides an excellent method for calculating the effect aerosol scattering has on light direction and polarization. However, dust and biomass burning aerosols can be complex mixtures with non-spherical shapes. Manfred et al.

(2018) and Espinosa et al. (2019) have both shown that a spherical approximation of biomass burning aerosol is sometimes inaccurate. Dust is another light-absorbing, aspherical, and atmospherically important species whose optical properties have been shown to be poorly quantified and thus contribute significantly to uncertainty in the global radiative balance (Xie et al., 2017;Schuster et al., 2016). For these species, more computationally expensive approximations (e.g. T-matrix, Rayleigh-Debye-Gans, or discrete dipole approximation) may need to be used to calculate the aerosol scattering matrix, $\overline{\overline{P(\theta)}}$ (Liu and Mishchenko,

2018;Bohren and Huffman, 1983). *In situ* optical, microphysical, and polarimetric measurements of these complex aerosols are necessary to evaluate these models, upon which remote sensing retrievals of aerosols are dependent (Schuster et al., 2019;Mishchenko et al., 2007).

Various instruments have been used in the past to measure the directional scattering of light *in situ*. An excellent review of earlier methods is given in Bohren and Huffman (1983). Here, we focus on the latest techniques to provide context for our own instrument.

The Polarized Imaging Nephelometer, PI-Neph, was developed as an aircraft instrument for measuring the directionality and polarization of light scattering (Dolgos and Martins, 2014). It uses a wide-angle lens and a folded laser path. Light scattering at three wavelengths (473 nm, 532 nm, and 671 nm) can be sequentially interrogated in two different input laser linear polarizations. The scattered light is imaged using a cooled charge-coupled detector (CCD) which provides excellent sensitivity. This sensitivity means that the instrument is capable of measuring scattering from submicron particles like biomass burning aerosol, but also is

sensitive to stray light in the instrument sample volume. This stray light introduces noise into the measurement and is minimized by incorporating a large sample cell (10 l), allowing the stray light to be dispersed and absorbed by the black interior rather than reflecting into the CCD. While increasing the sample cell volume decreases the stray light and thus increases precision, it also decreases the sample exchange rate, and therefore temporal resolution. This is especially important in aircraft measurements where airspeeds of 100-200 m s$^{-1}$ require fast response times (a few seconds) to achieve spatial resolutions <1 km. Another feature of the

PI-Neph is that it is operated within the aircraft cabin. This allows aerosol to be conditioned before being analyzed (e.g. controlling relative humidity, thermodenuding, or size selecting the aerosol via impactor.) The benefit of this mode of operation is it allows the quantitative selections of a portion of aerosol (e.g. PM1) for investigation, but it does increase the complexity of comparing measurements with remote sensors. Remote sensing techniques measure light scattering by aerosol at ambient relative humidity and temperature, which likely affects composition via partitioning and water uptake.

For a more direct comparison of *in situ* and remote measurements, the Open Imaging Nephelometer (OI-Neph) was developed (Espinosa, 2017). The OI-Neph is a wing-mounted probe operated at a single wavelength (532 nm) that was designed to maintain alignment despite the physical movement of the wing in flight. This allows angularly-resolved radiance and polarimetry measurements of aerosol at ambient relative humidity (RH) and temperature. This also means that the OI-Neph measures the phase function from all ambient aerosol, as opposed to in-cabin instruments that are unable to fully sample the coarse mode due to inertial

losses in inlets. Another recent instrument is a commercial laser imaging nephelometer, LiNeph, from Air Photon (Baltimore, MD, USA).

This original LiNeph, described in Manfred et al. (2018), was designed to investigate the optical properties at near-ultraviolet wavelengths, equipped with lasers at 375 nm and 405 nm. This instrument uses circularly polarized light, and thus only measures the directionality of the scattered light, with no information regarding changes in polarity. Nonetheless, Manfred et al. showed that

lab-generated biomass burning particles did not scatter light in a manner that was consistent with Mie theory, which was likely due to the irregular shape and composition of the particles. Manfred et al. also showed that the optical properties of biomass burning aerosol varied from fire-to-fire and also after evaporation by a thermodenuder.

**Table 1. Comparison of some existing imaging nephelometers.**

| Instrument name | PI-Neph | OI-Neph | LiNeph (Manfred et al. 2018) | LiNeph (This work) |
|---|---|---|---|---|
| Wavelength(s) (nm) | 473, 532, 671 | 532 | 375, 405 | 405, 660 |
| Scattering matrix elements measured | $P_{11}$ and $P_{12}$ | $P_{11}$ and $P_{12}$ | $P_{11}$ | $P_{11}$ and $P_{12}$ |
| Aerosol sample exchange rate | 30 sec | Instantaneous | 40-60 sec | <13 sec |
| Aerosol pre-conditioning | Yes | None | Yes | Yes |

Here, we present scientific results from an improved Laser Imaging Nephelometer. This instrument incorporates design elements from both the PI-Neph and the LiNeph of Manfred et al. (2018), but is optimized for the rapidly changing aerosol conditions as one might encounter on an aircraft. Table 1 shows a comparison of the four instruments. The LiNeph is operated inside the aircraft cabin, and thus the aerosol sample can be conditioned to a controlled temperature and relative humidity; this design also enables it to operate at ground sites. The instrument sample cell was designed to minimize sample volume and the duty cycle of the

instrument was doubled by arranging the laser beams parallel to each other (see Fig. 1a). This allows the beams to be imaged simultaneously by the cameras. In contrast, a coaxial laser alignment meant they needed to be viewed sequentially by alternating which laser was on. The new LiNeph also has the added capability of measuring the scattering matrix element $P_{12}$, like the PI-Neph (Dolgos and Martins, 2014.) The PI-Neph achieves this by changing the polarization the laser using a liquid crystal variable retarder. By rotating the laser polarization to be roughly parallel, and then perpendicular, to the optical axis of the wide angle lens,

one can calculate $P_{12}$ from the scattered light measurements. For the LiNeph, we achieve similar orientations of the optical axis of the wide angle lens to the laser polarization by using two detectors. One is placed such that the optical axis of the wide angle lens is roughly parallel to the incident laser polarization, and the other is roughly perpendicular to the laser polarization, as shown in Fig. 1b. This allows us to measure the scattered light in the two orientations required for deriving $P_{12}$, simultaneously.

We selected two visible wavelengths (405 and 660 nm) to be recorded with each image, which allows for ready comparison with

the NOAA AOP instrument suite (Langridge et al., 2011;Lack et al., 2012). We use two wide-angle lenses and cooled CCDs to collect images of light scattered perpendicular and parallel to the lasers' polarization, allowing us to measure both the directionality and the polarization of light scattered by the sample.

## 2 Instrument description and methods

### 2.1 Theory

To describe the scattered light measured by the instruments above, we use Stokes's formalism

$$\begin{pmatrix} I_{sca}(\theta) \\ Q_{sca}(\theta) \\ U_{sca}(\theta) \\ V_{sca}(\theta) \end{pmatrix} = \frac{\sigma_{sca} \cdot \Delta V}{4\pi r^2} \cdot \overline{\overline{P(\theta)}} \cdot \begin{pmatrix} I_{in} \\ Q_{in} \\ U_{in} \\ V_{in} \end{pmatrix} \qquad (1).$$

Here, the incident and scattered light are described using Stokes's parameters for intensity (I) and the polarization ellipse (Q, U, and V)(Hansen and Travis, 1974). $r$ is the distance of the detector from the scattering event. When interpreting this equation, it is helpful to remember that the total scattering, i.e. integrated over all angles, should be equal to the product of the scattering

coefficient ($\sigma_{sca}$), the volume of the scattering medium ($\Delta V$), and the incident light intensity ($I_{in}$). Thus, it becomes clear that the aerosol scattering matrix, $\overline{\overline{P(\theta)}}$, is a) the only factor with an angular dependence and b) normalized such that it will integrate over all angles to equal $4\pi$. We can think of the aerosol scattering matrix as a function which evaluates the probability that incident light will be scattered in a given direction, while preserving information regarding its polarization. $\overline{\overline{P(\theta)}}$, defined in Eqn. (2), is a 4x4 matrix which due to symmetry consists of six unique elements for randomly oriented particles that do not possess intrinsic optical activity (Bohren and Huffman, 1983).

$$\overline{\overline{P(\theta)}} = \begin{pmatrix} P_{11}(\theta) & P_{12}(\theta) & 0 & 0 \\ P_{12}(\theta) & P_{22}(\theta) & 0 & 0 \\ 0 & 0 & P_{33}(\theta) & P_{34}(\theta) \\ 0 & 0 & -P_{34}(\theta) & P_{44}(\theta) \end{pmatrix} \tag{2}$$

Using different approximation methods, each of these elements can be calculated for a particle of known size and composition. Under single-scatter conditions, the elements of an aerosol population are the scattering cross section-weighted sum of the elements from individual particles. Mie theory is the most commonly used method for calculating the intensity and polarization state of light after scattering with spherical aerosols, and thus is the foundation of aerosol microphysical retrievals (Dubovik and King, 2000;Mie, 1908). For the LiNeph, the incident light can be defined with respect to the orientation of the observing camera relative to the polarization of the linearly polarized lasers. For the perpendicular ("Perp") camera shown in Fig. 1, the Stokes vector used to evaluate Eq. (3) is:

$$\begin{pmatrix} I_{in} \\ Q_{in} \\ U_{in} \\ V_{in} \end{pmatrix}_{Perp} = \begin{pmatrix} 1 \\ -1 \\ 0 \\ 0 \end{pmatrix} \tag{3}$$

This is because the axis of the CCD (along the x-axis) is approximately orthogonal to the polarization of the lasers (along the z-axis.) In reality, a small offset in the z-axis (~3.6 mm) from the optical axis of the wide angle lens introduces a small angle which describes the scattering plane rotation angle, $\eta$ (Dolgos, 2014). For now, we assume $\eta$ is zero, although we revisit this assumption in Section 2.4 as it has important implications for the accuracy of the measurement. Solving Eq. 1 for this idealized case means that the measured parameter, $I_{scat,Perp}(\theta)$ contains information about two elements from the scattering matrix, $P_{11}(\theta)$ and $P_{12}(\theta)$, as shown in Eq. (4):

$$I_{sca,Perp}(\theta) = \frac{\sigma_{sca} \cdot \Delta V}{2\pi r^2} \cdot [P_{11}(\theta) - P_{12}(\theta)] \tag{4}$$

Similar treatment for the parallel ("Para") camera shows that the combined measurements can be used to solve for both $P_{11}$ (commonly referred to as the scattering phase function) and $P_{12}$, as shown in Eq. (5) and (6):

$$\begin{pmatrix} I_{in} \\ Q_{in} \\ U_{in} \\ V_{in} \end{pmatrix}_{Para} = \begin{pmatrix} 1 \\ 1 \\ 0 \\ 0 \end{pmatrix} \tag{5}$$

$$I_{sca,Para}(\theta) = \frac{\sigma_{sca} \cdot \Delta V}{2\pi r^2} \cdot [P_{11}(\theta) + P_{12}(\theta)] \tag{6}$$

$P_{12}$ is typically reported for convenience as the degree of linear polarization (DoLP), $-P_{12}/P_{11}$. Below we will discuss the capabilities and limitations of the new aircraft-deployable LiNeph, as well as present some initial data from the FIREX-AQ field campaign studying wildfire smoke onboard the NASA DC-8.

## 2.2 Instrument design and operation

The LiNeph uses two continuous wave laser beams as the light sources to measure the light scattering of an aerosol sample at two different wavelengths. The emissions from the OBIS 660 nm LX 100 mW diode laser (Coherent, Santa Clara CA, USA) and the

LuxX 405-120 diode laser (Omicron, Rodgau, Germany) are directed into the aerosol sample chamber using turning mirrors, shown in Fig. 1a. For all the work presented here, the lasers were operated at 15% of full power, 15 mW and 18 mW for the 660 nm and 405 nm lasers, respectively. Before entering the aerosol sample chamber, the lasers pass through a Glan-Taylor polarizer (GT10-A, Thorlabs, Newton MA, USA) to ensure linear polarization and then an anti-reflective coated window (VPW42-A, Thorlabs, Newton MA, USA). We use a series of four black nylon 3D-printed apertures to reduce stray light entering the chamber. The stray light reflects off the interior of the black-painted sample cell and is imaged by the detectors, resulting in increased noise when there is low signal. The lasers have a diagonal offset which enables aerosol scattering from both beams to be imaged by both cameras, as shown in Fig. 1b.

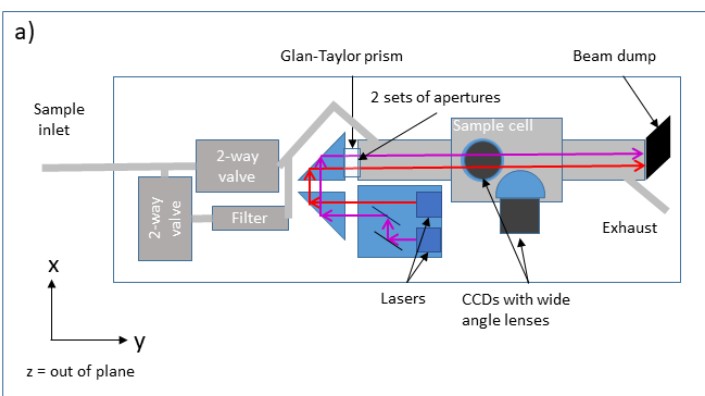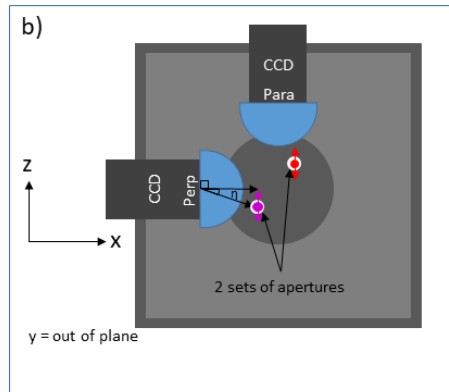

**Figure 1. Geometry of laser imaging nephelometer. a) Simplified schematic showing the sample flow and laser paths. b) Schematic of the aerosol sample cell indicating the optical geometry of the wide-angle lenses and both lasers, including the scattering plane rotation angle (η). The cameras are identified by their orientation relative to the laser polarization, either Parallel or Perpendicular.**

Sample flow is pulled through the instrument sample cell by an external diaphragm pump and controlled by a mass flow controller (MCR-50, Alicat, Tuscon AZ, USA). For the FIREX-AQ mission aboard the NASA DC-8, a sample flowrate of 15 l min$^{-1}$ was used to maximize the sample exchange rate in the ~3 l sample volume, and thus improve the ability of the instrument to resolve spatial changes in aerosol concentration as the aircraft penetrated a smoke plume. For ground-based measurements, lower flow rates could be used if the aerosol composition is not expected to change rapidly. Since some particles may be hygroscopic, the instrument exhaust is characterized using a temperature and relative humidity probe (HMP110, Vaisala, Vantaa, Finland). The sample cell pressure is monitored using a precision pressure transducer (PPT0015AXN5VA, Honeywell, Charlotte SC, USA).

The two LiNeph CCD detector arrays (16-bit, 2750x2200 pixel, cooled to -40 °C, Trius-SX694, Starlight Xpress, Bracknell, UK) record the images from the orthogonally mounted wide-angle, f-theta type lenses (FE185C046HA-1; Fujifilm, Tokyo, Japan). These images show the light scattered by everything in the field of view of the wide-angle lenses, including: the instrument optics and interior, gases with non-negligible scattering cross-sections, and particles. Since the particles are the species of interest, a high efficiency particulate arrestance (HEPA) filter was interposed upstream of the sample volume to remove particles approximately every five minutes for a 45 second duration; see Fig. 1a. The two-way valves (MDM-060DT, Hanbay Inc., Virginia Beach VA, USA) were automated and controlled using a custom Labview program (National Instruments, Austin TX, USA) that also handled the data acquisition.

The particle-free, background images with identical optical and detector conditions (laser power and CCD exposure time) are averaged from before and after a sample period, and the resulting image is subtracted from sample images. An example background-subtracted image is shown in Fig. 2. The two arcs are from particles illuminated by the 405 nm and 660 nm lasers on the top and bottom, respectively, distorted by the wide-angle lens. The reported units are bits, which shows the full scale of the 16-bit detector. Bits are converted into a differential scattering coefficient (σ°), Mm$^{-1}$ sr$^{-1}$, which will be a function of the CCD exposure time, as described in Section 2.3. In this image, the lasers propagate from left to right, and thus lower (higher) pixel columns show

forward (backward) scattering. In addition to light scattered directly by the particles, the CCD arrays also detect stray or multiply scattered light. An example of multiply scattered light is shown in Fig. 2. Columns 20-60 and rows 60-100 show the light scattered by the particles and then again by the other wide-angle lens. Our background subtraction cannot account for these secondary scattering events, but we minimize the effect by darkening the interior of the instrument where possible and by excluding the affected pixels from the analysis.

From the background-subtracted image, two Gaussian functions are fit to each pixel column, one for each laser, excluding parts of the image that don't overlap with the laser path to the extent that is possible. The area under these Gaussian fits proportional to the particle scattering matrix elements ($P_{11}+P_{12}$) and ($P_{11}-P_{12}$) for the cameras oriented parallel and perpendicular to laser polarization, respectively. The precision of this method of analysis depends on the temporal stability of both the detector and the subtracted elements, which in turn rely on the stability of the pressure of the sample and the power of the lasers. If any of these elements, or

the detector, changed in sensitivity, then the background subtracted images would be biased. During FIREX-AQ, we accounted for varying sample pressure by taking filter samples before and after any changes in aircraft altitude. Since automated filter samples are collected every five minutes, barring operator deferment, we will show in Sect. 3.1 that no drift in detector response was detected for greater than 10 minutes. Further, filter periods at the same altitude during the research flights (>2 hours) showed similar response, indicating that the lasers and detectors are stable and that the filtered air images are valid representations of

instrument background and scattered light from gaseous species.

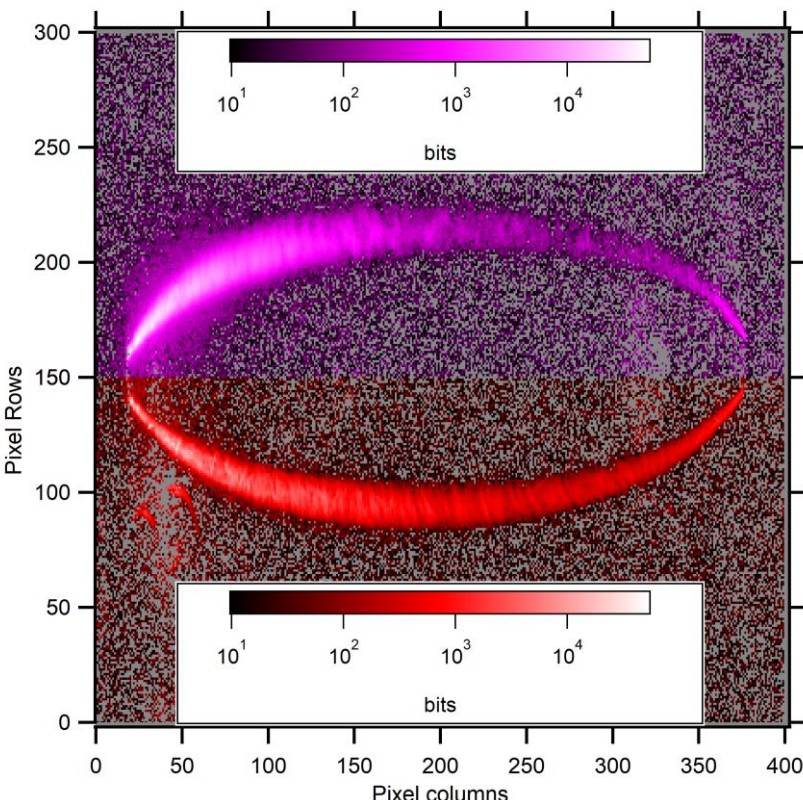

**Figure 2. Colorized image of particle light scattering from room air. Individual pixel values in bits are the difference between an aerosol scattering image (light scattered by particles, gases, and the instrument itself) and a filter image (gases and the instrument itself.) Two different logarithmic color scales are used to illustrate the scattering from the 405 nm laser (magenta, top) and 660 nm laser (red, bottom.)**
**Curvature of the laser profile is due to the extremely wide-angle (fisheye) camera lens. In some cases the subtraction of noise can result in a small negative value, which is shown as grey.**

**2.3 Aerosol generation and conditioning**

The calibration of the LiNeph requires the sampling of gases and aerosols of known size and composition. Supplementary Fig. S1 shows the lab set up for calibration of the LiNeph. For calibrations using a pure gas, either $CO_2$ or He, the LiNeph is pumped down to 125 hPa using an IDP3 scroll pump (Agilent, Santa Clara CA, USA), and backfilled with the gas of choice to ambient pressure. To ensure complete flushing of the sample volume, this process is repeated three times before a "He only" or "$CO_2$ only" measurement is made. For aerosol measurements, the sample diaphragm pump is disconnected and a nebulizer is used to generate 2 l min$^{-1}$ of positive-pressure flow containing particles of known size and refractive index through the instrument. To ensure a consistent sample throughout the instrument volume, we verify that the aerosol loading and RH have been constant for least 5 minutes before beginning to take a measurement for calibration.

Here, we also present some data from the 2019 FIREX-AQ aircraft campaign. Figure 3 shows a schematic of the aerosol sampling, pre-conditioning, and measurement components aboard the NASA DC-8. During this campaign the LiNeph was mounted adjacent to integrating nephelometers (TSI Model 3563, Shoreview MN, USA) from the NASA Langley Aerosol Research Group (LARGE). All the instruments discussed here sampled from the LARGE/University of Hawaii aerosol isokinetic inlet which has a geometric diameter upper cut size of 4-5 µm (McNaughton et al., 2007;Chen et al., 2011). The LARGE group also operated a Laser Aerosol Spectrometer (model 3340A, TSI, Shoreview MN, USA) from the aerosol inlet, and measured the dry particle size distribution at a frequency of 1 Hz (Moore et al., 2021). The aerosol sampled by the LiNeph was also dried to less than 20% RH using Nafion driers and passed through a cyclone with a calculated cut size of 1.5 µm aerodynamic diameter. The aircraft cabin was temperature controlled, resulting in average sampling temperatures of ~26 °C, although temperatures could rise as high as 36 °C when sampling at low altitudes for an extended duration. A dry scroll pump (TriScroll 300, Agilent, Santa Clara CA, USA) provided vacuum for both the integrating nephelometers and the LiNeph. The flow controller specific to the LiNeph controlled the flow to 15 volumetric l min$^{-1}$. We will compare the integrated scattering measured by the LiNeph with the scattering derived from the measurements from the NOAA Aerosol Optical Properties (AOP) instrument suite, which includes cavity ringdown spectrometers (CRDS) and photoacoustic aerosol spectrometers (PAS) at wavelengths of 405, 532, and 664 nm (Langridge et al., 2011;Lack et al., 2012). The AOP instrument package sampled from the same aircraft inlet as the LiNeph and LARGE nephelometers, but located less than 2 m away. These measurements of aerosol extinction and absorption can be used to calculate the integrated aerosol scattering at the wavelengths interrogated by the LiNeph.

There are two important differences between the aerosol measured by the LiNeph and the AOP instrument suite. Firstly, the AOP uses an impactor to remove dry aerosols with aerodynamic diameters >2.5 µm, while the LiNeph cyclone cut point is 1.5 µm. However, in smoke plumes the difference between the total light scattering of PM1.5 vs PM2.5 is negligible due to the overwhelming abundance of submicron particles. This was confirmed using size distributions from the LAS, which showed few particles with diameters greater than 1 µm (Moore et al., 2021). Secondly, during high particle concentrations that were common in the sampled smoke plumes, the aerosol sampled by the CRDS needed to be diluted. Without the dilution system, the uncertainty of the CRDS extinction measurement for dry scattering coefficients >100 Mm$^{-1}$ is ± 5%, but with the dilution system in line, the uncertainty is estimated to be ± 30%. This added uncertainty was characterized in the field and may be due to incomplete mixing of the filtered and unfiltered sample flows.

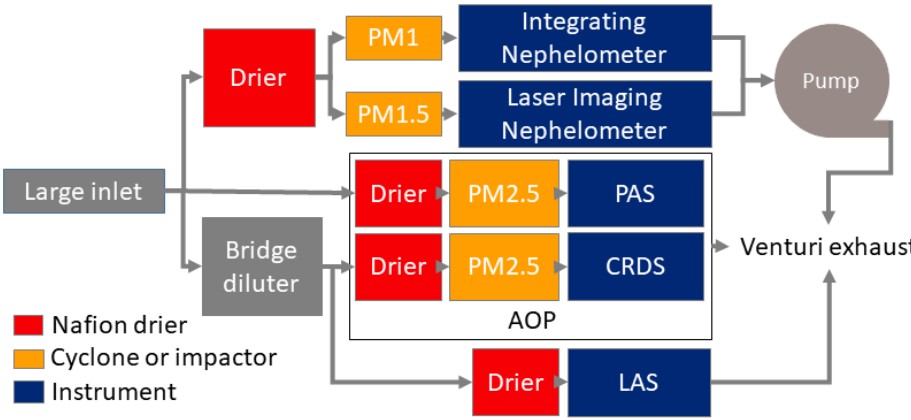

**Figure 3. Partial diagram of aerosol sampling suite aboard NASA DC-8 during FIREX-AQ.**

## 2.4 Calibration and data reduction

We convert each LiNeph image into σ° (scattering intensity as a function of scattering angle, Mm$^{-1}$ sr$^{-1}$) by applying two calibrations. σ° measured by each CCD array (see equations 4 and 6) can then be used to solve for the normalized scattering matrix elements of interest, $P_{11}$ and $P_{12}$.

First, we convert the pixel intensity to σ° (Mm$^{-1}$ sr$^{-1}$) by comparing the area under a Gaussian fit at a given scattering angle (i.e. pixel column) to the theoretical scattering of particle-free air or $CO_2$, both of which are well described by Rayleigh scattering

(Manfred et al., 2018;Dolgos and Martins, 2014). Second, we establish which pixel column corresponds to which scattering angle by identifying local maxima and minima observed in the measured phase function of NIST-traceable polystyrene latex spheres (PSLs). Since PSLs are well characterized with respect to size, dispersion, shape, and refractive index, we can calculate the expected scattering matrix elements with a high degree of confidence. Figure 4 shows the good agreement between the measured and calculated $P_{11}$ and DoLP. Mie theory is used to calculate the expected phase function and degree of linear polarization for two

theoretical instrument geometries. Although the optical axis of each lens is parallel or perpendicular to the polarization of the laser, the beams themselves are offset from the optical axis by ~3.6 mm. This is roughly defined by the geometry of the series of four concentric apertures through which the lasers are introduced into the sample cell. Due to the nature of the wide angle lens, this can result in a scattering plane rotation angle, η, shown in Fig.1 and described in detail by Dolgos et al. (2014.) Figure 4 shows predictions of what the expected P11 and DoLP would be if η is assumed to be zero (gold line) or if an estimated η were used (teal

line). We do not observe improved agreement between the observed and calculated values for $P_{11}$ nor DoLP by using the estimated η, thus we treat our estimated η as an upper limit. The estimated upper limit of η varies with scattering angle and does not exceed 40°; see Supplementary Fig. S2-S5. For some scattering angles (e.g. near 30° and 135°), there is stray light from the instrument background which introduces additional noise as extraneous features.

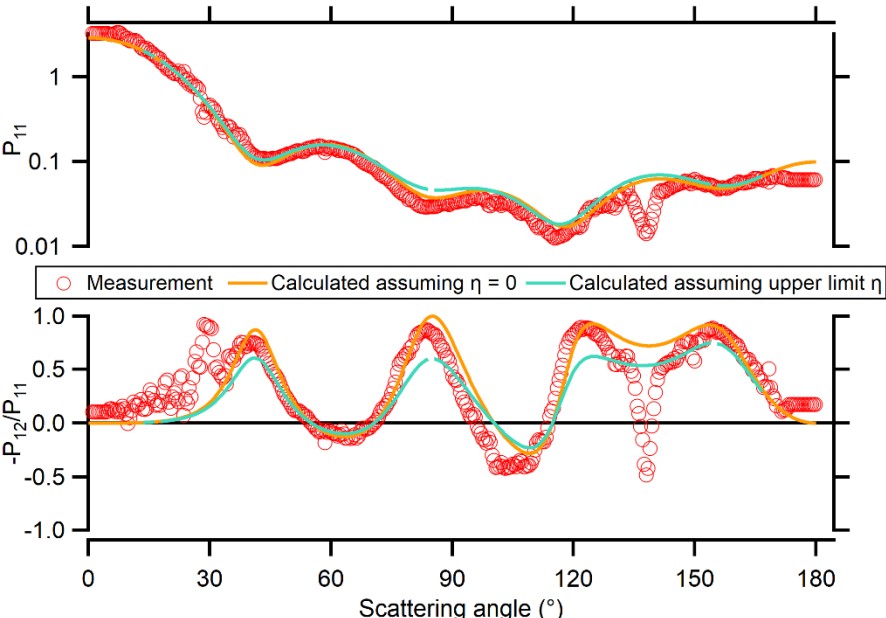

Figure 4. Phase function ($P_{11}$, top) and degree of linear polarization ($-P_{12}/P_{11}$, bottom) for $\lambda = 660$ nm light scattered by $D_p = 900$ nm PSLs. Measurements are shown as red circles and the values calculated from Mie theory are shown as solid lines. The gold line shows the values calculated if scattering plane rotation ($\eta$) is assumed to be zero. If we use an upper estimate of $\eta$, Mie theory predicts that the observations would follow the teal line. Stray light from the inside of the instrument introduces noise at scattering angles around 30° and 135° for both data products.

The conversion from pixel intensity (bits) to $\sigma°$ (Mm$^{-1}$ sr$^{-1}$), referred to as the differential scattering calibration in this work, also accounts for two types of image distortion. The first is distortion by the wide-angle lens, where the image of the area illuminated by the laser beam appears wider (is projected onto more CCD pixels) close to a scattering angle of 90° (see Fig. 2). The second distortion is due to the varying scattering path length for a given scattering angle. That is, the length of the volume of air defined by the laser and the ~0.5° pixel viewing angle is shortest close to 90°. Thus, the differential scattering calibration can have a small effect on the angular calibration by slightly shifting the pixel location of the PSL phase function local minima and maxima. The feature shifts are small and become negligible with just one iteration of the pixel column-to-angle and differential scattering calibration analyses. The PSL angular calibration is shown in Supplementary Fig. S6, where dots indicate the raw, initial fitting, and the open circles symbols indicate the final calibration. Error bars on the calibrated data indicate the 95% confidence interval of the linear regression (scattering angle as a function of pixel column) that is the angular calibration. For each calibration point, we calculated the 95% confidence interval by the propagating the variance associated with the linear regression slope and intercept while accounting for covariance between the slope and intercept. The average 95% confidence interval for the 14 points is 0.9 ± 0.2°.

We determine the differential scattering coefficient calibration by measuring $\sigma°$ of $CO_2$ using circularly polarized light for each laser and each detector for a single alignment geometry. Circularly polarized light is necessary for this portion of the calibration because for linearly polarized light scattered in the Rayleigh regime, $P_{11}(\theta) + P_{12}(\theta)$ (the scattering intensity observed by the Para CCD assuming $\eta = 0°$) at 90° is extremely small and hard to measure accurately. For circularly polarized light, the light scattering at 90° is ½ of the scattering at 0°, as shown in Fig. 5.

Circular polarization was achieved by placing a zero-order quarter-wave plate (WPQSM05-405/670, Thorlabs, Newton MA, USA) after the Glan-Taylor prism for each laser. To remove the background signal from stray light scattering off the interior of the instrument, we purged the sample volume and backfilled with helium, which has a negligible scattering cross section. After subtracting the illuminated helium image from the illuminated $CO_2$ image, we applied the same fitting protocol used for measuring aerosol scattering. The area under each of these Gaussian fits is shown as red circles in Fig. 5. The red circles are the uncalibrated

σ° for 935 hPa of CO₂ illuminated by 660 nm light as viewed by the "Parallel" CCD (see Fig. 1). The black dotted line shows σ° calculated using Mie theory, the measured sample pressure, and the scattering cross section of $CO_2$ from Penndorf (1957). The ratio of the theoretical σ° ($Mm^{-1}$ $sr^{-1}$) to the raw σ° (bits) is the differential scattering coefficient calibration, shown as orange triangles. A 6$^{th}$-order polynomial fit is used to smooth and extrapolate the calibration function between the smallest (7°) and largest (171°) measured scattering angles. This calibration is applied to all raw data to correct for the lens distortion and varying path length as described above, as well as correct for differences in CCD array sensitivity. This allows the measured σ° to be compared and therefore isolate the scattering matrix elements of interest.

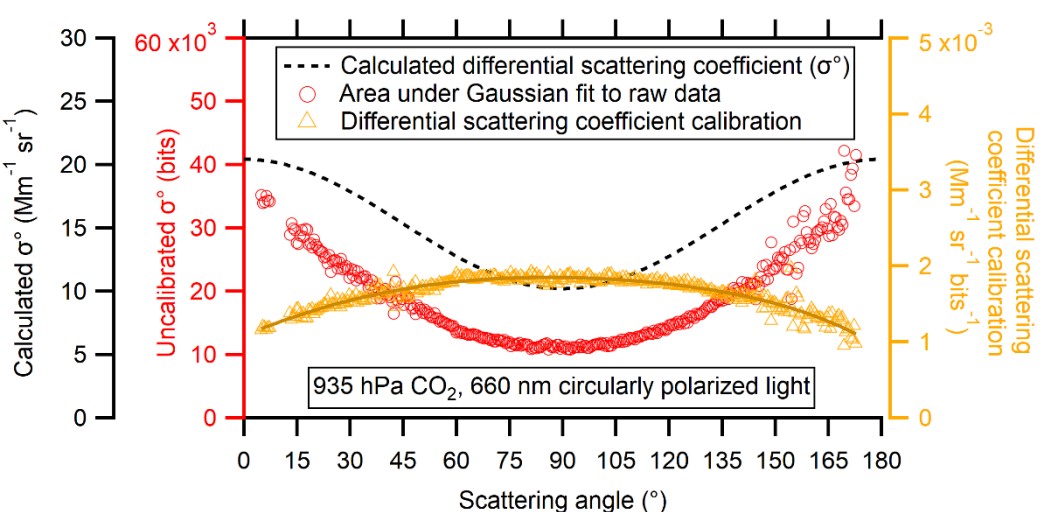

**Figure 5. Differential scattering coefficient calibration for 935 hPa of CO₂ using circularly polarized, 660 nm light. The differential scattering coefficient calibration (dark orange line) is a 6$^{th}$-order polynomial fit to the ratio of calculated σ° (black dotted line, Mm$^{-1}$ sr$^{-1}$) to the area of the Gaussian fit to the background-subtracted image (orange triangles, bits.)**

## 3 Operation and performance

### 3.1 Precision and accuracy

The precision of the LiNeph depends on both the stability of the instrument (laser power and detector response) and also the homogeneity of light-scattering entities in the sampled air. We can evaluate the stability of the instrument response in the lab first by ensuring homogeneity of the sampled air – that is by using particle-free CO₂. Over the duration of ten minutes, we observed no statistically significant change in light scattered by pure CO₂ as measured by our CCD arrays, suggesting that over that time period there is no significant drift in either the sensitivity CCD arrays or the laser output. Further, no correlation ($R^2 = 0.007$) was observed when comparing the light scattered at different angles, see Supplementary Fig. S7, indicating that the noise observed in the measurement was due to either the Gaussian peak fitting routine and/or electronic noise within the CCD array. If the variation were due to laser power fluctuations, then the observed light scattering intensity at different angles would have been correlated.

Another potential source of instability in the instrument is mechanical vibration induced by the aircraft. The instrument uses a rigid optical cage system (30 mm cage components, Thorlabs, Newton MA, USA) to minimize susceptibility to mechanical perturbations. The cage system, the laser platform, and the sample cell are all mounted to a modified aluminum U-channel (1630T45, McMaster-Carr, Sante Fe Springs CA, USA) that provides excellent rigidity. No deviation in the alignment was observed during the FIREX-AQ campaign as measured by the pixel position of the Gaussian fit maximum value.

We accomplish two things by using the area under a Gaussian fit of the signals in the pixel rows of the CCD at a given angle (pixel column) as a measure of scattered light. Firstly, we account for lens distortion of the laser beam diameter. Secondly, we effectively average electronic noise observed in individual pixels; i.e. there is less noise in the measured Gaussian fits of a measurement than

there would be measuring just the pixel intensity at the peak of the signal. For example, multiple measurements of a peak pixel at a given scattering angle (e.g., 104°) may have a relative standard deviation of 54%, but the area under the Gaussian fit for the same angle has a relative standard deviation of 22%. One might expect similar accuracy improvement by summing the pixels containing the laser, as is the case with the PI-Neph (Dolgos and Martins, 2014.) Theoretically, the difference between the two methods should be small. One benefit of the Gaussian fit technique is that it may be less sensitive to which pixel rows one designates as representing the light scattered by the aerosol. The Gaussian fit technique can also readily account for a changing baseline (e.g. if multiple scattering illuminates the inside of the instrument.) However, by summing the signal, one is less sensitive to the inhomogeneous background of the instrument (see Fig. 2).

## 3.2 Limit of quantification and laser attenuation

The limit of quantification is defined for each scattering angle measured. We conservatively define the minimum signal required for quantification of aerosol scattering to be a Gaussian fit with an amplitude that is at least ten times greater than the noise (one standard deviation) measured in the background subtraction sample. Noise may result from either electronic noise in the CCD array or from light reflecting off the interior of the instrument body. This means that the noise varies spatially in each image, e.g. more noise is observed in forward scattering directions due to window glow where the lasers enter the sample cavity. Supplementary Figure S8 shows that the noise in the Gaussian fit area observed at each scattering angle is ~2% of the signal or 200 bits, whichever is larger, for a 0.5 s exposure time. This means that increasing the CCD exposure time can allow measurements of the phase function when total scattering is low (e.g. $\sigma_{scat}$ is ~14 Mm$^{-1}$ for the differential scattering calibration in Fig. 5.) Varying the exposure duration can also be useful for phase functions that are strongly forward scattering, and thus require a broad dynamic range. If the aerosol population is unchanging, two sets of differential scattering functions can be measured and then combined: one with a short exposure (to capture intense forward scattering without saturating the CCD) and one with a long exposure (to increase the signal-to-noise for less intense backscattering angles.)

It is also conceivable that very high aerosol concentrations could attenuate the propagating laser, thereby biasing the observed scattering to the forward scattering angles. However, even with an extinction coefficient of 10,000 Mm$^{-1}$, the laser would only be attenuated at most by 0.7%. The maximum scattering coefficient observed during FIREX-AQ, within intense smoke plumes, was 8,000 Mm$^{-1}$ and thus we consider this to be a minor source of error.

High aerosol concentrations can also affect the accuracy of $\sigma°$ measurements because of multiple scattering. In this instance a photon is scattered by a particle in a direction consistent with $\sigma°$, but is scattered by a second particle before being detected. Gogoi et al. (2009) showed that there was a small but measureable multiple scattering effect (reduction in radiance measured) when the optical depth was greater than 0.01. The maximum distance from scattering entity to the detector in the LiNeph is ~36 cm (for particles scattering in the forward direction), resulting in an optical depth of 0.028 at the highest observed integrated scattering coefficient. This suggests that for the higher concentrations (scattering coefficient greater than ~3,333 Mm$^{-1}$), there may be a small negative bias. Monte Carlo radiative transfer simulations by Ge et al. (2011) show that for a field-of-view of 1° and with particle diameters of 500 nm, the negative bias will be less than 3%.

## 4 Field Measurements

### 4.1 Uncertainty due to aerosol sample inhomogeneity

Having addressed the inherent instrument uncertainties, we will now analyze the uncertainties associated with specific measurement environments and samples. There is a concern that, due to the large volume of the sample chamber of the LiNeph,

there might not be a homogeneous sample illuminated by the lasers. It is important that each observed solid scattering angle

contains a representative distribution of the aerosol. One instance where this would not be the case is if rare but highly scattering particles transit through the sample cell but only transect the laser at a few angles. This would result in spikes observed in the recorded phase function. This was not observed during FIREX-AQ because the aerosol distribution was dominated by very high concentrations of small particles, and because a $PM_{1.5}$ cyclone removed larger dust and ash particles that may have been present. This was verified by the size distribution measurements with impactors with an even larger cut size, $PM_{2.5}$.

Another potential source of error during ambient measurements is a rapidly changing $\beta_{scat}$, e.g. when there is an increase in the aerosol number concentration. If the sampled $\beta_{scat}$ increases rapidly, the $\beta_{scat}$ gradient within the sample cell will be observed in the scattering phase function. For example, if the $\beta_{scat}$ at the instrument inlet increases during a measurement, there may be more scattering in the section of the sample cell corresponding to forward scattering angles than in the portion of the cell corresponding to backscattering. To minimize this effect for the sampling of wildfire plumes, the LiNeph was designed with a minimal internal

volume, albeit at the expense of increased background noise due to stray light. During FIREX-AQ, the LiNeph was operated at 15 l min$^{-1}$, which means an aerosol exchange rate of less than 12 s for the approximately 3 l sample cell if perfect mixing is assumed. Supplementary Fig. S9 shows the change in total measured CCD signal (no Gaussian fits or image processing) while measuring well-mixed smoke and interposing a HEPA filter. Imposing the filter at t = 0 s results in the removal of smoke particles and leaves only the light-scattering gases. An exponential fit shows a 2.6 second time constant, which suggests that the sample cell should not

be characterized as a well-mixed reactor. Plug or laminar flow through the center of the instrument may result in a functionally faster aerosol exchange rate. This allows for transition periods from background air to smoke plume air to be minimized. Additionally, we report angularly-resolved radiance and polarimetric measurements only when the prior measurement of the integrated scattering is within 15% of the current measurement, usually about 2.5 s later, indicating that we are not likely in a transition period that would skew the phase function shape. This criteria is equally important when merging long and short exposure

radiance measurements to capture strong forward scattering and weak back or side scattering, as described Section 3.2.

For the FIREX-AQ mission, we sampled smoke plumes aboard the NASA DC-8 traveling 159 ± 6 m s$^{-1}$ while sampling smoke. This means that aerosol composition would change rapidly as we entered and exited the ~44 km wide smoke plume. We will show that the LiNeph had sufficient temporal resolution to capture the larger features of smoke plumes by comparing the integrated scattering measured by the LiNeph at 0.24 Hz with the integrated scattering calculated from 1 Hz measurements of extinction and

absorption by the AOP instrument suite (Langridge et al., 2011). If the sample exchange rate was insufficient, the integrated scattering measured by the LiNeph will appear as a moving average of the AOP-derived scattering coefficient. Panel a) of Fig. 6 shows that there is sufficient aerosol exchange to capture the major features in a large smoke plume, although the finer details are lost. The calculated Pearson correlation coefficient between the two measurements is 0.96.

Panel b) of Fig. 6 expands this analysis by showing the integrated scattering for 6 (5) FIREX-AQ research flights, at 405 (660)

nm. The nearest neighbors approach was used to account for truncated angles. Flights were excluded that were missing data from the AOP measurements. A linear fit to the remaining data points show 21% (2%) more scattering measured by the LiNeph than the AOP-derived scattering at 405 (660) nm, with an $R^2$ = 0.99 (0.97). While this is within the specified accuracy of the AOP-derived scattering measurement for diluted samples, an analysis of undiluted measurements (Supplementary Fig. S10) shows less variance in the measurements, $R^2$ = 0.99 (0.98), and shows a consistently 30% (24%) higher scattering measured by the LiNeph

versus the AOP-derived scattering measurements, for 405 (660) nm. If we take the AOP-derived scattering as a truth measurement, i.e. without its own error, we can say that the LiNeph is precise within <2%, although with a positive bias of ~30%, likely due to calibration error. This is consistent with the reported accuracy of similar techniques, i.e. the PI-Neph reports agreement with commercial integrating nephelometers to within 5% (Espinosa et al., 2017).

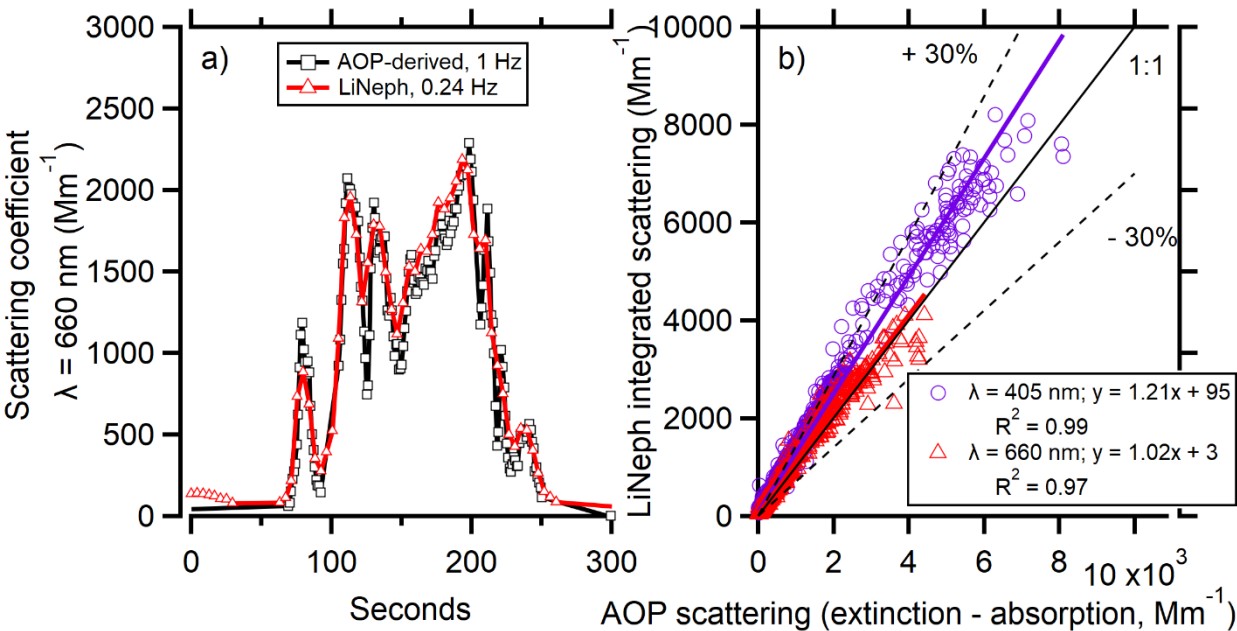

Figure 6. a) shows the time series of aerosol scattering at 660 nm measured by the LiNeph (red triangles) and calculated from the AOP suite measurements (black squares). The Pearson correlation coefficient is 0.96. b) The integrated scattering measured by the LiNeph as a function of the scattering derived from the AOP measurements. Red triangles and purple circles show measurements at 660 and 405 nm, respectively, along with linear regression fits, shown as solid lines. A 1:1 line and the ±30% bounds are shown as black solid line and two dash lines, respectively.

**4.2 Example angularly-resolved radiance and polarimetry measurements of smoke**

Having established that the LiNeph was likely measuring a homogeneous sample of smoke, we can now investigate the directionality and polarization of light scattered by wildfire smoke. Figure 7 shows the angularly-resolved radiance and polarimetry measurements at 405 nm of smoke during two transects of the Williams Flats fire plume on August 7th, 2019. This fire was initiated by a lightning strike and consumed over 44,000 acres of fuel including timber, short grass, light slash from logging, and a coniferous overstory over 25 days (InciWeb). The fire emitted an intense smoke plume extending downwind over 104 km and up to 44 km wide. The traces in Fig. 7 show the mean plus two standard deviations of two sets of measurements. Each set of measurements is from a single transect perpendicular to the axis of the smoke plume. While spacing of plume transects during FIREX-AQ were intended to produce pseudo-Lagrangian data, in fact the aircraft frequently traveled downwind at a rate faster than the plume advection (ratio of smoke age to elapsed time during all of FIREX-AQ was 0.8-6.4 as reported in Supplementary Fig. S3 of Wiggins et al. (2020)). For the August 7th flight, this ratio was about 3. Smoke age was estimated using wind speed and distance of the measurement from the fire. Smoke in Transect 1 was emitted approximately 1 hour prior to being sampled, and smoke in the second transect considered here, Transect 10, was emitted approximately 4.4 hours prior to being sampled. Panel a) of Fig. 7 shows that there is a significant difference in the directional scattering of 405 nm light by smoke, although panel b) shows that the change in linear polarization as a function of scattering angle appears to be consistent between the two plumes. The change in directional scattering was likely due in part to the change in mean particle size between the two transects. Supplementary Fig. S11 shows the average normalized number-weighted size distributions for both transects as measured by the LAS with an applied ammonium sulfate calibration (Moore et al., 2021). The mode diameter of Transect 1 was 174 nm while Transect 10 was 225 nm, only 51 nm larger.

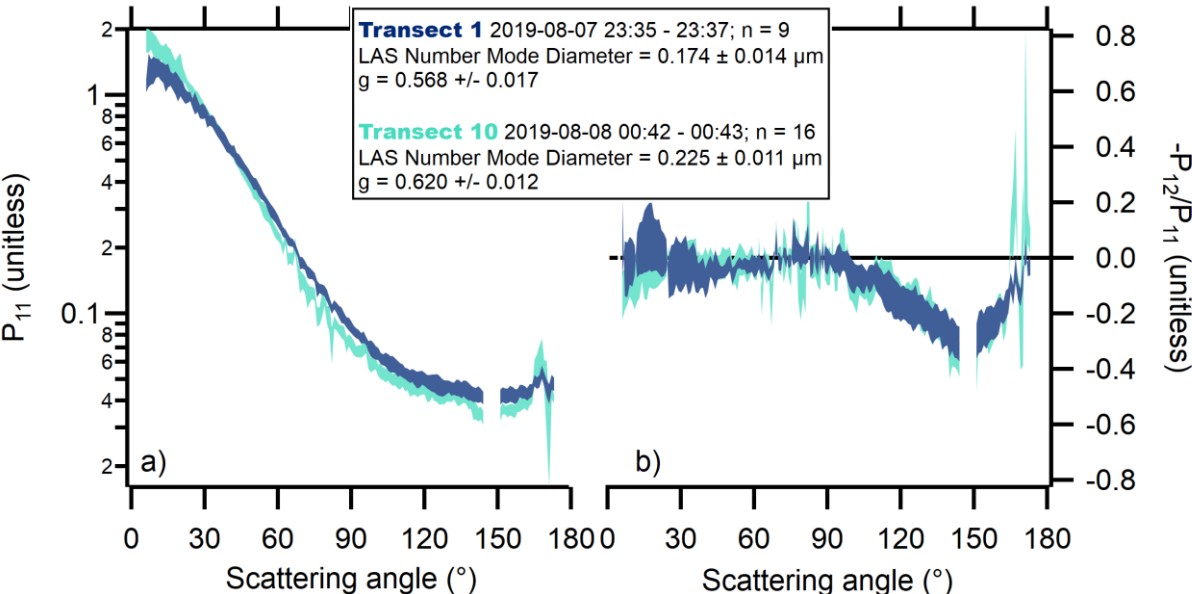

**Figure 7. Angularly-resolved radiance and polarimetry measurements of wildfire smoke. Panel a) shows the phase function ($P_{11}$) and panel b) shows the degree of linear polarization (DoLP, $-P_{12}/P_{11}$).**

This modest growth could have been caused by changing emissions and/or atmospheric processing. The asymmetry parameter, described below, increases from 0.568 to 0.620, showing an increase in forward scattering that is consistent with increasing particle size. However, it is important to note that changes in particle composition, hence refractive index, have also been observed as a consequence of photochemical aging in biomass burning aerosol. The degree of linear polarization provides additional information that may be useful in determining to what degree changing refractive index and size account for the changing phase function. Analysis of the smoke optical properties and their changes with plume will be the subject of future study and is beyond the scope of this work.

**4.3 Direct measurement of the asymmetry parameter**

One important application of phase function measurements is the calculation of the asymmetry parameter. The asymmetry parameter, g, is the intensity-weighted cosine average of the scattering angle (Andrews et al., 2006). It is calculated following Eq. (7):

$$g = \frac{1}{2} \int_0^\pi \cos(\theta)\, P_{11}(\theta)\, \sin(\theta)\, d\theta \qquad (7).$$

The asymmetry parameter is used as a computationally efficient way to approximate the fraction of light that is scattered into the upper hemisphere, or up-scatter fraction, in radiative transfer models (Wiscombe and Grams, 1976). Despite its importance in understanding the direct radiative effect of aerosols in models, the asymmetry parameter is rarely measured directly. Instead, it is commonly estimated from hemispheric backscatter measurements by integrating nephelometers or else calculated from Mie theory (Andrews et al., 2006;Moosmüller and Ogren, 2017). Unfortunately, Marshall et al. (1995) used Mie theory to show that the typical method of estimating using hemispheric backscatter measurements will overestimate the asymmetry parameter for accumulation mode aerosols. Further, whether or not Mie theory is appropriate for predicting biomass burning aerosol phase functions is an area of active research (Manfred et al., 2018;Liu and Mishchenko, 2018).

Future work will explore the relationship between the asymmetry parameter and the hemispheric backscatter fraction, both of which can be derived from the phase function directly. It will also be of interest, although beyond the scope of this work, to evaluate whether Mie theory can be used, along with the particle size distribution measurements and assumed refractive indices, to predict the hemispheric backscattering measured by the integrated nephelometers and the LiNeph.

To calculate g, we first used a nearest-neighbors method to account for truncation, i.e. inability to measure scattering at the extreme forward and backwards ($\theta < 7°$ or $\theta > 171°$) angles. We investigated the effect of truncation on the asymmetry parameter using simulated phase functions calculated from measured particle size distributions during FIREX-AQ. We found that truncation affected the asymmetry parameter by less than 1% due to the small particle size. Truncation will likely have a larger effect for

particle size distributions with supermicron particles and therefore strong forward scattering.

The phase function measurements (e.g. Fig. 7) allow for precise measurements of the asymmetry parameter with a relative standard deviation of less than 3%. But, as discussed in Section 2.4, the geometry of the LiNeph requires that the lasers be offset from the optical axis of the wide angle lenses, introducing a non-zero $\eta$. Based on measurements of PSLs (Fig. S2-S5), we can set an upper bound of this effect on the measured $\sigma°$. We used the same method to investigate the effect of $\eta$ on the measured asymmetry

parameter. Supplementary Figure S12 shows that for polydisperse lognormal aerosol size distributions with a mode around 200 nm, with varying refractive indices, the effect of a non-zero $\eta$ on g is a bias of less than 2%. For the largest modeled aerosol population, with a mode at 400 nm, the bias was 5%. This effect is small in part because the geometry of the instrument results in offsetting biases when $\sigma°_{\text{"Perp"}}$ and $\sigma°_{\text{"Para"}}$ are combined to calculate $P_{11}$. For the four $P_{11}$ measurements of PSL calculated from the data in Supplemental Fig. S2-S5, the average ratio of measured to Mie-calculated g was $1.01 \pm 0.09$.

**5 Conclusion**

We present here a new instrument, the LiNeph, for the simultaneous measurement of two scattering matrix elements, $P_{11}$ and $P_{12}$, at two wavelengths. We have described in detail the data processing required to convert the three-dimensional raw images into two-dimensional $\sigma°$ that are the sum and difference of two scattering matrix elements. From these two $\sigma°$, we can solve for the individual scattering matrix elements, and also calculate the asymmetry parameter, g. We described the iterative calibration process

that makes combining these vectors possible. We validated our method by showing good agreement with Mie theory for spherical particles of known composition in the lab.

We also investigated two potential sources of error relating to the $\sigma°$ measurements. First, we quantified the inherent instrument precision by measuring the variability of Gaussian fits in the presence of a homogeneous sample, pure $CO_2$. The standard deviation for an individual row of pixels (~0.5° scattering angle) was the larger of 2% of the signal or 200 bits for a 0.5 s exposure time.

Second, we investigated the potential for sample inhomogeneity to influence $\sigma°$ measurements during the FIREX-AQ campaign specifically. The good temporal agreement between the $\sigma_{\text{scat}}$ measured by the LiNeph and 1-Hz optical instruments suggest that there was sufficient temporal resolution to capture major trends in aerosol concentration gradients and that there was a statistically representative sample at each measured scattering angle.

Finally, we showed that $\sigma°$ measurements were sufficiently precise to identify changes in the phase function resulting from at least

a 51 nm growth in particle diameter, although additional contributions from changes in refractive index cannot be ruled out. Additional work is required to evaluate whether Mie theory or the more morphologically rigorous T-matrix method is appropriate for reproducing the measured phase function and polarization of light scattered by smoke.

Finally, we showed that we can precisely (less than 3% relative standard deviation) and accurately (within 10% for the PSLs examined in this work) determine the asymmetry parameter. Direct determinations of the asymmetry parameter, as opposed to

derivation from the measured hemispheric backscatter fraction using a priori assumptions, are uncommon. Future work will focus on evaluating the relationship between measurements of the asymmetry parameter and the hemispheric backscatter fraction, and understanding the implications of the measured asymmetry parameter on the direct radiative effect caused by fresh wildfire smoke.

*Data availability*. The FIREX-AQ field campaign data are publicly available in the NASA Airborne Science Data for Atmospheric

Composition Archive at https://doi.org/10.5067/SUBORBITAL/FIREXAQ2019/DATA001 (FIREX-AQ Science Team, 2021).

*Author contributions*. ATA wrote the first draft of the manuscript and performed the laboratory experiments. ATA, NLW, and DMM conceived of and designed the instrument modifications. ATA and FE built the instrument. ATA, NLW, CAB, ML, KS, RHM, and EBW made the FIREX-AQ airborne measurements. ATA, NLW, CAB, ML, KS, RHM, EBW, and DMM contributed

to the data interpretation and manuscript revisions.

*Competing interests*. The authors declare that they have no conflict of interest.

*Acknowledgements*. We thank the FIREX-AQ project scientists Jim Crawford, Shuka Schwarz, Carsten Warneke, and Jack Dibb,

as well as the pilots and crew of the NASA DC-8.

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
