# Peer review of "Laser Imaging Nephelometer for aircraft deployment"

_Atmospheric Measurement Techniques, 2021_

## Author Comment (AC1)

We thank both reviewers for their thorough reviews. We appreciate their constructive comments and suggestions and the manuscript has been revised accordingly. Our point-by-point responses to the comments are presented below. The comments are in **black**, responses are in **blue**, and revised manuscript are in red with specific changes marked by underline.
* * *
The manuscript describes a new version of the LiNeph polar nephelometer that has increased stability as well as sensitivity to the second element of the aerosol scattering matrix, P12. Details of instrument design and calibration are presented, followed by example measurements of smoke plumes from the FIREX-AQ campaign.

The instrument provides airborne measurements of P11 and P12 which are currently very limited but have high potential value, especially to future remote sensing missions the will be flying advanced polarimeters like PACE and ACCP. The new LiNeph design represents several improvements over past nephelometers, particularly the addition of a second camera allows two roughly orthogonal input polarization states to be measured simultaneously which improves measurement accuracy and time resolution. Moreover, an important potential source of error, inhomogeneity in the sample volume, is discussed in much more depth than in previous studies involving imaging nephelometers. The work is therefore novel, has the potential for significant impact and is clearly relevant to AMT. There are, however, a few minor points that I feel must be addressed before the article can be published, most notably the magnitude and impact of the deviation from ideal input polarization states needs to be discussed.

Thank you for the kind comments. We endeavor to address your points of concern below.

**GENERAL COMMENTS:**

(A) In order to measure a range of scattering angles, the direction at which the chief ray enters the camera (and potentially the orientation of the scattering plane) must vary across the FOV (i.e., the lens cannot be object-space telecentric). Therefore, given an axial symmetric lens, it is geometrically impossible for the direction of linear polarization to be exactly perpendicular to the scattering plane in one camera and parallel in the other for both (not spatially coincident) lasers simultaneously. How far are the actual laser polarization orientations from the ideal orientations represented by Equations (3) and (5) in Section 2.1? Figure 1 implies potentially large deviations from the ideal case, but perhaps this schematic is not scale. Regardless, this issue must be addressed in the text as even small misalignments can lead to relatively large biases (see Dolgos and Martins, 2014).

We appreciate the reviewer pointing out this important consideration and address it more fully in the manuscript and supplementary materials, as described below. In short, we consulted Dolgos and Martins (2014) as well as the PhD dissertation by Gergely Dolgos (2014) to first estimate scattering plane rotation (η) caused by the non-ideal alignment of the observed scattering plane with respect to the polarization of the incident laser. We found that agreement between the measured signal and the Mie calculated differential scattering coefficients were not markedly improved by including our estimate of η. Thus we use our estimated η as an upper limit of what the potential effects of a non-zero η might be for the differential scattering coefficient and the derived asymmetry parameter.

Line 114:

This is because the radially symmetric axis of the camera (along the x-axis) is approximately orthogonal to the polarization of the lasers (along the z-axis.) In reality, a small offset in the z-axis (~3.6 mm) from the radial center of the wide angle lens introduces a small angle which describes the scattering plane rotation angle, η (Dolgos, 2014). For now, we assume η is zero, although we revisit this assumption in Section 2.4 as it has important implications for the accuracy of the measurement. Solving Eq. 1 for this idealized case means that the measured parameter, $I_{scat,Perp}(\theta)$ contains information about two elements from the scattering matrix, $P_{11}(\theta)$ and $P_{12}(\theta)$, as shown in Eq. (4):

Line 228:

Figure 4 shows the good agreement between the measured and calculated $P_{11}$ and DoLP. Mie theory is used to calculate the expected phase function and degree of linear polarization for two theoretical instrument geometries. Although the center axis of each camera is parallel or perpendicular to the polarization of the laser, the beams themselves are offset from the center axis by ~3.6 mm. This is roughly defined by the geometry of the series of four concentric apertures through which the lasers are introduced into the sample cell. Due to the nature of the wide angle lens, this can result in a scattering plane rotation angle, η, shown in Fig.1 and described in detail by Dolgos et al. (2014.) Figure 4 shows predictions of what the expected $P_{11}$ and DoLP would be if η is assumed to be zero (gold line) or if an estimated η were used (teal line). We do not observe improved agreement between the observed $P_{11}$ nor DoLP by using the estimated η, thus we treat our estimated η as an upper limit (see Supplementary Fig S2-S5.) For some scattering angles (e.g. near 30° and 135°), there is stray light from the instrument background which introduces additional noise as extraneous features.

Line 448:

The phase function measurements (e.g. Fig. 7) allow for precise measurements of the asymmetry parameter with a relative standard deviation of less than 3%. But, as discussed in Section 2.4, the geometry of the LiNeph requires that the lasers be offset from the optical axis of the wide angle lenses, introducing a non-zero η. Based on measurements of PSLs (Fig. S2-S5), we can set an upper bound of this effect on the measured σ°. We used the same method to investigate the effect of η on the measured asymmetry parameter. Supplementary Figure S12 shows that for polydisperse lognormal aerosol size distributions a mode around 200 nm, with varying refractive indices, the effect of a non-zero η on g is a bias of less than 2%. For the largest modeled aerosol population, with a mode at 400 nm, the bias was 5%. This effect is small in part because the geometry of the instrument results in offsetting biases when $\sigma°_{"Perp"}$ and $\sigma°_{"Para"}$ are combined to calculate $P_{11}$.

 (B) There is no direct validation of the new P11 and -P12/P11 products. This is generally understandable given the lack of polar nephelometer measurements available for intercomparison and the challenges associated with modeling scattering matrix elements of natural aerosol. However, in other polar nephelometers, observations of artificial PSL spheres have frequently been leveraged to gain a better understanding of instrument performance. The PSL measurements here cannot be used as a completely wholistic validation since they are also used in instrument calibration but examining PSL P11 and -P12/P11 could still help to increase confidence in (and help better understand the accuracy of) the

measurement. This is especially true for the amplitude of the peaks and valleys of the measured scattering matrix elements, which are only very moderately affected by the scattering angular calibration. Moreover, imperfect input polarization states will strongly impact the amplitude of the PSL features, particular in -P12/P11, so good agreement with Mie theory there could be one way to confirm that slight misalignment of the laser polarization (as discussed in (A) above) is not a cause for significant measurement degradation. In my view, Figure 4 needs to be augmented (or possibly replaced) with a figure showing the processed P11 and -P12/P11 PSL data and corresponding Mie calculations.

We have modified to Figure 4 to show the $P_{11}$ and $-P_{12}/P_{11}$ for PSL data, as described in response to the previous comment. However, we have maintained the convention of presenting the differential scattering coefficients for each detector in the Supplementary Fig. S2-S5. We feel that this is a more straightforward way to evaluate the potential influence of η, given each detector will have a different η(θ).

**DETAILED COMMENTS:**

1) LN 21: The range of scattering intensities that can be accurately measured is actually never mentioned in the text (only in the abstract). How was this 50-80,000 Mm-1 estimate derived? On a related note, the Rayleigh scattering coefficient of the CO2 used in calibration is much less than 50 Mm-1, especially in the red channel. Thus, I am wondering if it is possible to measure at loadings less 50 Mm-1 under stable laboratory conditions (like those used in the CO2 calibration)?

The scattering coefficient range of 50-8,000 $Mm^{-1}$ (80,000 is a typo, see Fig. 6) describes the range of scattering coefficients where a phase function was measured during the FIREX-AQ field campaign. Longer exposure times (up to 5 s) are used in the lab for the $CO_2$ calibrations, which was indeed at a much lower scattering coefficient as described in Section 3.2, line 300.

This means that increasing the CCD exposure time can allow measurements of the phase function when total scattering is low (e.g. $\sigma_{scat}$ is ~14 $Mm^{-1}$ for the differential scattering calibration in Fig. 5.) Varying the exposure duration can also be useful for phase functions that are strongly forward scattering, and thus require a broad dynamic range. If the aerosol population is unchanging, two sets of differential scattering functions can be measured and then combined: one with a short exposure (to capture intense forward scattering without saturating the CCD) and one with a long exposure (to increase the signal-to-noise for less intense backscattering angles.)

2) LN 29: I'm having trouble following this sentence. Why is the observational geometry only important at certain distances? And I'm not clear specifically what "characteristics of the scattering entities" the authors are referring to. Please clarify.

We have clarified as follows:

It is important to account for observational geometry when retrieving aerosol microphysical and optical properties from scattered light measurements.

3) LN 38: It should be clarified that Mie theory requires knowledge of particle size in addition to composition.

The text has been modified as per your suggestion.

For spherical aerosols of known size and composition, Mie theory provides an excellent method for calculating the effect aerosol scattering has on light direction and polarization.

4)   LN 52: It would be good to note that these are two different input laser linear polarizations (as opposed to analyzers on the detector/camera end).

The text has been modified as per your suggestion.

It uses a wide-angle lens and a folded laser path. Light scattering at three wavelengths (473 nm, 532 nm, and 671 nm) can be sequentially interrogated in two different input laser linear polarizations.

5)   LN 56: Upper and lower case "L" is used to represent liter at different points in the text. The abbreviation used should be consistent.

This has been corrected on line 56.

6)   LN 69: It might be a little unfair to say cabin-based instruments cannot sample particles bigger than ~1µm. (Later in the text it is stated that the inlet used here has a cutoff around 4-5µm.)

We had originally intended that clause to define "the coarse mode," but have removed it for clarity.

This also means that the OI-Neph measures the phase function from all ambient aerosol, as opposed to in-cabin instruments that are unable to fully sample the coarse mode, ~~particles with a diameter greater than ~1 µm,~~ due to inertial losses in inlets.

7)   LN 85: I had a bit of trouble following the remaining sentences in this section and sometimes found it difficult to tell if the orthogonal orientations mentioned were in reference to the polarization state of the lasers or the optical axes of the cameras. The text should be adjusted for clarity and made more precise. A new figure (or even reference to figure 1) might also help convey the geometric details of the different designs.

We have elaborated on the geometry of the new instrument and included a reference to Fig. 1.

The instrument sample cell is designed to minimize sample volume and the duty cycle of the instrument is doubled by arranging the laser beams parallel to each other (see Fig. 1a). This allows the beams to be imaged simultaneously by the CCDs. In contrast, a coaxial laser alignment meant they needed to be viewed sequentially by alternating which laser was on. The new LiNeph also has the added capability of measuring the scattering matrix element P$_{12}$, like the PI-Neph (Dolgos and Martins, 2014.) The PI-Neph achieves this by changing the polarization the laser using a liquid crystal variable retarder. By rotating the laser polarization to be roughly parallel, and then perpendicular, to the optical axis of the wide angle lens, one can calculate P$_{12}$ from the scattered light measurements. For the LiNeph, we achieve similar orientations of the optical axis of the wide angle lens to the laser polarization by using two detectors. One is placed such that the optical axis of the wide angle lens is roughly parallel to the incident laser polarization, and the other is roughly perpendicular to the laser polarization, as shown in Fig. 1b. This allows us to measure the scattered light in the two orientations required for deriving P$_{12}$, simultaneously.

8)   LN 100: 4πr^2 would mean that the normalization is dependent on the distance between the observation and scattering event. Is this what is intended here, or is the normalization such that the integral of P11 over all angles equals 4π? Also, the definition of r should be included in the text.

We have included the definition of *r* and agree that the integral of $P_{11}$ should be $4\pi$. We have modified the manuscript as follows:

Here, the incident and scattered light are described using Stokes's parameters for intensity (I) and the polarization ellipse (Q, U, and V)(Hansen and Travis, 1974). *r is the distance of the detector from the scattering event.* When interpreting this equation, it is helpful to remember that the total scattering, i.e. integrated over all angles, should be equal to the product of the scattering coefficient ($\sigma_{sca}$), the volume of the scattering medium ($\Delta V$), and the incident light intensity ($I_{in}$). Thus, it becomes clear that the aerosol scattering matrix, $\overline{\overline{P(\theta)}}$, is a) the only factor with an angular dependence and b) normalized such that it will integrate over all angles to equal $4\pi$.

9)    LN 103: Random orientation alone is not completely sufficient to guarantee only six unique scattering matrix elements (e.g., see Chapter 4 of Mishchenko et al. 2002). The scope of the statement should be appropriately narrowed.

We have modified the text as follows:

We can think of the aerosol scattering matrix as a function which evaluates the probability that incident light will be scattered in a given direction, while preserving information regarding its polarization. $\overline{\overline{P(\theta)}}$, defined in Eqn. (2), is a 4x4 matrix which due to symmetry consists of six unique elements for randomly oriented particles that do not possess intrinsic optical activity, e.g. biogenic sugars (Bohren and Huffman, 1983).

10)    LN 106: It should be clarified that this sum must be weighted by the scattering cross section of the individual particles. (The normalized scattering matrix elements do not account for differences in the total, angularly integrated scattering between particles).

We have modified the text as follows:

Under single-scatter conditions, the elements of an aerosol population are  the scattering cross section-weighted sum of the elements from individual particles.

11)    LN 108: The year of the Mie reference should be 1908.

This has been corrected.

12)    LN 109: I'm struggling to follow this sentence. It would be good to clearly define coordinate system in which the Stokes vectors that follow are written.

We have clarified as follows:

This is because the η axis of the CCD (along the x-axis) is approximately orthogonal to the polarization of the lasers (along the z-axis.) In reality, a small offset in the z-axis (~3.6 mm) from the optical axis of the wide angle lens introduces a small angle which describes the scattering plane rotation angle, η (Dolgos, 2014). For now, we assume η is zero, although we revisit this assumption in Section 2.4 as it has important implications for the accuracy of the measurement. Solving Eq. 1 for this idealized case means that the measured parameter, $I_{scat,Perp}(\theta)$ contains information about two elements from the scattering matrix, $P_{11}(\theta)$ and $P_{12}(\theta)$, as shown in Eq. (4):

13)    Figure 1: Would it be possible to add the location of the 3D-printed apertures to this figure?

We have modified Figure 1 and included the linear offset of the apertures (~3.6 mm) relative to the optical axis of the wide angle lens to the text.

Although the optical axis of each camera is parallel or perpendicular to the fast-axis of the laser, the beams themselves are offset from the optical axis by ~3.6 mm, as defined by the geometry of the series of apertures through which the lasers are introduced into the sample cell.

14) LN 138: It would be helpful to state the volume of the LiNeph sample cell earlier in the text so that the reader can better contextualize the flow rate presented here.

We have modified the text as follows:

For the FIREX-AQ mission aboard the NASA DC-8, a sample flowrate of 15 l min$^{-1}$ was used to maximize the sample exchange rate in the ~3 l sample volume, and thus improve the ability of the instrument to resolve spatial changes in aerosol concentration as the aircraft penetrated a smoke plume.

15) LN 159: Could the authors double check the columns specified? The main stray light feature I see is centered below pixel 50.

We have measured the digital image of the feature and modified the text.

Columns 20-60 and rows 60-100 show the light scattered by the particles and then again by the other wide-angle lens.

16) LN 215: I would usually refer to this as a 2-D matrix (the values of intensity specified over the two dimensions of rows and columns). Although, in some sense it is 3-D because polarization state (i.e., camera) could also be considered a dimension. Either way, I would rephrase this sentence to avoid confusion.

We have removed the statement defining the number of dimensions of the matrix to avoid confusion.

We convert each LiNeph image into σ° (scattering intensity as a function of scattering angle, Mm$^{-1}$ sr$^{-1}$) by applying two calibrations.

17) LN 248: I think perhaps the authors mean the signal in the "para" camera goes to zero at θ=90°? The P11 element of the scattering matrix never goes to zero for Rayleigh scattering. Also, it might be more accurate to say the signal becomes very weak since the Rayleigh depolarization correction will prevent it from going completely to zero (see below).

The text has been modifies as follows:

Circularly polarized light is necessary for this portion of the calibration because for linearly polarized light scattered in the Rayleigh regime, $P_{11}(\theta)+P_{12}(\theta)$ (the scattering intensity observed by the Para camera assuming η = 0°) at 90° is extremely small and hard to measure accurately. For circularly polarized light, the light scattering at 90° is ½ of the scattering at 0°, as shown in Fig. 5.

18) LN 257: Was a Rayleigh depolarization correction (i.e., see Eq 2.15-16 of Hansen and Travis, 1974) used here? The effect can be relatively large for CO2 (Young, 1980).

We thank the reviewer for bringing the work of Young to our attention. For the depolarization correction, we used the values in Penndorf (1957) from Gucker and Basu (1953.) In comparison, the

calculated scattering coefficient only increases by 1.7% when using a $\rho^t$ of 0.0805 (Gucker and Basu, 1953) versus the more accurate 0.0708 (Young, 1980.)

19)    LN 258: Why is this unit bit^2? Isn't the value just the sum of bits in all pixels at a given angle?

Upon reflection, this area under the Gaussian fit should be in units of bits*pixel column, where column is dimensionless. We have modified this throughout the text and figures.

20)    LN 276: Was this test performed in the lab? In an aircraft context, is there a possibility of slight changes in mechanical alignment altering the calibration, especially where the laser passes through the 3D-printed apertures? I would suggest adding a short discussion of potential impacts of mechanical vibrations on calibration.

We have added the following:

Another potential source of instability in the instrument is mechanical vibration induced by the aircraft. The instrument uses a rigid optical cage system (30 mm cage components, Thorlabs, Newton MA, USA) to minimize susceptibility to mechanical perturbations. The cage system, the laser platform, and the sample cell are all mounted to a modified 24.5 cm-wide aluminum U-channel (1630T45, McMaster-Carr, Sante Fe Springs CA, USA) that provides excellent rigidity. No deviation in the alignment was observed during the FIREX-AQ campaign as measured by the pixel position of the Gaussian fit maximum value.

21)    LN 277: The advantages of performing a Gaussian fit over using the intensity at the peak of the signal are made clear here. A third approach that has been used in other imaging nephelometers (e.g., the PI-Neph) is to take the sum of the counts of all pixels containing the beam in each column. Perhaps there are strengths and weaknesses to both approaches, and it would be interesting if the authors could provide some discussion of the motivation for their particular Gaussian fit based approach.

We have added the following discussion:

One might expect similar accuracy improvement by summing the pixels containing the laser, as is the case with the PI-Neph (Dolgos and Martins, 2014.) Theoretically, the difference between the two methods should be small. The benefit of the Gaussian fit technique is that it is less sensitive to the precision with which one defines the pixels that contain the laser and the Gaussian fit can readily account for a changing baseline (e.g. multiple scattering illuminating the inside of the instrument.) However, by summing the signal, one is less sensitive to the inhomogeneous background of the instrument (see Fig. 2).

22)    LN 297: It might also be worth noting that multiple scattering inside the chamber also has the potential to bias the measurement (e.g., see Gogoi et al., 2009), although that effect is also likely small at the concentrations sampled here.

We have included the following based on Dolgos (thesis):

High aerosol concentrations can also affect the accuracy of σ° measurements because of multiple scattering. In this instance a photon is scattered by a particle in a direction consistent with σ°, but is scattered by a second particle before being detected. Gogoi et al. (2009) showed that there was a small but measureable multiple scattering effect (reduction in radiance measured) when the optical depth was greater than 0.01. The maximum distance from scattering entity to the detector in the LiNeph is ~36

cm (for particles scattering in the forward direction), resulting in an optical depth of 0.028 at the highest observed integrated scattering coefficient. This suggests that for the higher concentrations (scattering coefficient greater than ~3,333 Mm$^{-1}$), there may be a small negative bias. Monte Carlo radiative transfer simulations by Ge et al. (2011) show that for a field-of-view of 1° and with particle diameters of 500 nm, the negative bias will be less than 3%.

23)  LN 299: The abstract says the measurement is valid to 80,000 Mm-1, which would presumably produce close to 8x0.7%=~6% attention. The values in the abstract should be backed-up by and made consistent with the corresponding discussion in the text.

There is a typo in the abstract, it should read 8,000 Mm$^{-1}$, consistent with the data shown in Fig. 6b. This has been corrected.

24)  LN 323: Does the use of the term "polarimetric measurements" mean that only P12 is filtered, or is the filter also applied to P11? This should be clarified in the text.

We have modified the text as follows:

Additionally, we report angularly-resolved radiance and polarimetric measurements only when the prior measurement of the integrated scattering is within 15% of the current measurement, usually about 2.5 s later, indicating that we are not likely in a transition period that would skew the phase function shape.

25)  LN 324: I would have expected this to be 1/0.24Hz = 4.2 seconds later. Why is it ~2.5 seconds?

The time between the individual measurements varies due to a) changing exposure times (between 10 and 2000 ms) and b) the slightly variable time to download the data from the CCD, approximately 2 s. For the data shown here, we combined a long (500 ms) and short (50 ms) exposure to get a more accurate representation of the scattered light, resulting in a merged phase function being reported every 4.2 seconds. We describe this starting on line 333:

Varying the exposure duration can also be useful for phase functions that are strongly forward scattering, and thus require a broad dynamic range. If the aerosol population is unchanging, two sets of differential scattering functions can be measured and then combined: one with a short exposure (to capture intense forward scattering without saturating the CCD) and one with a long exposure (to increase the signal-to-noise for less intense backscattering angles.)

And have added the following to line 371:

Additionally, we report angularly-resolved radiance and polarimetric measurements only when the prior measurement of the integrated scattering is within 15% of the current measurement, usually about 2.5 s later, indicating that we are not likely in a transition period that would skew the phase function shape. This criterion is equally important when merging long and short exposure radiance measurements to capture strong forward scattering and weak back or side scattering, as described Section 3.2.

26)  LN 334: The method used for handling the truncated phase function angles in the integral yielding total scattering should be specified. (Presumably the nearest neighbor approach that was used to calculate g?)

This has been added to line 376.

Panel b) of Fig. 6 expands this analysis by showing the integrated scattering for 6 (5) FIREX-AQ research flights, at 405 (660) nm. The nearest neighbors approach was used to account for truncated angles.

27) LN 397: Would it be possible to show some asymmetry parameter data? If not in the main text, then perhaps in the supplement? This would be especially interesting if it could be plotted against other estimates of g, or even just the backscattering fraction from the integrating nephelometer on an adjacent axis, but perhaps these comparisons are best saved for a later work.

We had initially intended to include those comparisons, but you have correctly anticipated that the scope of that comparison warrants a separate, more focused, publication. We discuss this on line 428:

Future work will explore the relationship between the asymmetry parameter and the hemispheric backscatter fraction, both of which can be derived from the phase function directly. It will also be of interest, although beyond the scope of this work, to evaluate whether Mie theory can be used, along with the particle size distribution measurements and assumed refractive indices, to predict the hemispheric backscattering measured by the integrated nephelometers and the LiNeph.

28) LN 412: The use of the word "polarimetry" here strikes me as a little misleading given that the observable differences were in P11.

We have reviewed the manuscript and now use "polarimetry" to describe only measurements of polarization. Instead we use either "angularly-resolved radiance measurements" to describe measurements of the differential scattering coefficient ($\sigma°$).

29) LN 416: This sentence needs to be softened, as only the precision of g was demonstrated to be better than 3% (not the accuracy).

We have modified the manuscript as follows:

Line 455:

The phase function measurements (e.g. Fig. 7) allow for precise measurements of the asymmetry parameter with a relative standard deviation of less than 3%. But, as discussed in Section 2.4, the geometry of the LiNeph requires that the lasers be offset from the optical axis of the wide angle lenses, introducing a non-zero η. Based on measurements of PSLs (Fig. S2-S5), we can set an upper bound of this effect on the measured $\sigma°$. We used the same method to investigate the effect of η on the measured asymmetry parameter. Supplementary Figure S12 shows that for polydisperse lognormal aerosol size distributions centered around 200 nm, with varying refractive indices, the effect of a non-zero η on g is a bias of less than 2%. For the largest modeled aerosol population, with a mode centered at 400 nm, the bias was 5%. This effect is small in part because the geometry of the instrument results in offsetting biases when $\sigma°_{"Perp"}$ and $\sigma°_{"Para"}$ are combined to calculate $P_{11}$. For the four $P_{11}$ measurements of PSL calculated from the data in Supplemental Fig. S2-S5, the average ratio of measured to Mie-calculated g was 1.01 ± 0.09.

Line 482:

Finally, we showed that we can precisely (less than 3% relative standard deviation) and accurately (within 10% for the PSLs examined in this work) determine the asymmetry parameter.

30) Figure S2: Which camera does this plot correspond to? Would it be possible to show data from both?

We have updated Supplemental Fig. S2 (now, S6) to include calibrations.

[Figure]

**Figure S6. Calibration for converting pixel column to scattering angle using local maxima and minima in σ° from PSLs as measured by a) "Perp" and b) "Para" cameras. Markers show maxima and minima from Mie theory as a function of the corresponding pixel column of the measured σ°, with the differential scattering coefficient calibration applied. Error bars indicate the 95% confidence interval from the linear fit. Dots show the pixel column where the maxima/minima would be located in the raw data, without the differential scattering coefficient calibration.**

31) Figure S4: The meaning of the black lines should be stated.

Caption has been updated.

**Figure S8. Standard deviation of the area under Gaussian fit as a function of the average area under the Gaussian fit for two exposure durations. Each symbol represents a scattering angle for a series of measurements. Black lines are linear regressions fits.**

**REFERENCES:**

Mishchenko, M.I., L.D. Travis, and A.A. Lacis, 2002: Scattering, Absorption, and Emission of Light by Small Particles. Cambridge University Press. https://pubs.giss.nasa.gov/abs/mi06300n.html

Dolgos, Gergely, and J. Vanderlei Martins. "Polarized Imaging Nephelometer for in situ airborne measurements of aerosol light scattering." Optics express 22.18 (2014): 21972-21990.

Hansen, James E., and Larry D. Travis. "Light scattering in planetary atmospheres." Space science reviews 16.4 (1974): 527-610.

Young, A. T. "Revised depolarization corrections for atmospheric extinction." Applied optics 19.20 (1980): 3427-3428.

Gogoi, Ankur, et al. "Detector array incorporated optical scattering instrument for nephelometric measurements on small particles." Measurement Science and Technology 20.9 (2009): 095901.

---

## Author Comment (AC2)

We thank both reviewers for their thorough reviews. We appreciate their constructive comments and suggestions and the manuscript has been revised accordingly. Our point-by-point responses to the comments are presented below. The comments are in **black**, responses are in **blue**, and revised manuscript are in red with specific changes marked by underline.
* * *
In this work, the setup and validation of a laser imaging nephelometer that can be used for aircraft deployment and measurement of polarization state of particles was introduced. In addition, this instrument was also applied in the FIREX-AQ campaign for measurement of smoke plume and was found to be able to measure very high scattering coefficients with high temporal resolution. This work expanded the application of LiNeph, which can play an important role in comprehensive measurement of aerosol optical properties. The manuscript fits well to the scope of AMT and I recommend it to be published after addressing the following comments listed below.

Thank you for positive review. Our responses to your comments are below.

Specific comment:

Although it is impossible to use polarized particles to calibrate PiNeph, it is helpful to check the measurement of -P11/P12 for non-polarized particle. Quantification of the uncertainty of P11/P12 measurement is very important for the measured polarization state of particles in field campaign.

We have modified to Figure 4 to show the $P_{11}$ and $-P_{12}/P_{11}$ for PSL data. However, we have maintained the convention of presenting the differential scattering coefficients for each detector in the Supplementary Fig. S2-S5. We feel that this is a more straightforward way to evaluate the potential influence of a change in the scattering plane rotation, η, given each detector will have a different η(θ).

Line 228:

Figure 4 shows the good agreement between the measured and calculated $P_{11}$ and DoLP. Mie theory is used to calculate the expected phase function and degree of linear polarization for two theoretical instrument geometries. Although the center axis of each camera is parallel or perpendicular to the polarization of the laser, the beams themselves are offset from the center axis by ~3.6 mm. This is roughly defined by the geometry of the series of four concentric apertures through which the lasers are introduced into the sample cell. Due to the nature of the wide angle lens, this can result in a scattering plane rotation angle, η, shown in Fig.1 and described in detail by Dolgos et al. (2014.) Figure 4 shows predictions of what the expected $P_{11}$ and DoLP would be if η is assumed to be zero (gold line) or if an estimated η were used (teal line). We do not observe improved agreement between the observed $P_{11}$ nor DoLP by using the estimated η, thus we treat our estimated η as an upper limit (see Supplementary Fig S2-S5.) For some scattering angles (e.g. near 30° and 135°), there is stray light from the instrument background which introduces additional noise as extraneous features.

Technical comments:

L138: It would be better to describe the cell volume and the exchange rate here.

We have added the sample cell volume to the text, but leave the exchange rate for the more detailed discussion later in the text.

Sample flow is pulled through the instrument sample cell by an external diaphragm pump and controlled by a mass flow controller (MCR-50, Alicat, Tuscon AZ, USA). For the FIREX-AQ mission aboard the NASA DC-8, a sample flowrate of 15 l min$^{-1}$ was used to maximize the sample exchange rate in the ~3 l sample volume, and thus improve the ability of the instrument to resolve spatial changes in aerosol concentration as the aircraft penetrated a smoke plume.

L142: Since you have measured the RH and temperature of instrument exhaust, how did they change during the measurement in the FIREX-AQ campaign?

We have included the following on line 211:

The aerosol sampled by the LiNeph was also dried to less than 20% RH using Nafion driers and passed through a cyclone with a calculated cut size of 1.5 µm aerodynamic diameter. The aircraft cabin was temperature controlled, resulting in average sampling temperatures of ~26 °C, although temperatures could rise as high as 36 °C when sampling at low altitudes for an extended duration.

L181: The calibration requires not only compositions but also the size of aerosol, as you mentioned later.

We have modified the text as follows:

The calibration of the LiNeph requires the sampling of gases and aerosols of known size and composition.

L195: The cut-off size impactors before each instruments were different from each other. Why don't you use impactors with the same cut-off size?

Although using the same cut-off size would have been better, we felt confident that for the in-plume sampling the aerosol scattering would be dominated by submicron particles. This was confirmed by measurements by the LAS which did not have an impactor. We have included the following in the text:

However, in smoke plumes the difference between the total light scattering of PM1.5 vs PM2.5 is negligible due to the overwhelming abundance of submicron particles. This was confirmed using size distributions from the LAS, which showed few particles with diameters greater than 1 µm (Moore et al., 2021).

L305: How large in specific do you mean by "very large ash particles"? Because there is a PM1.5 impactor before PiNeph, it seems that "very large ash particles" should be smaller than 1.5um.

We have removed the phrase "very large ash particles" from the text.

One instance where this would not be the case is if rare but highly scattering particles, for example very large ash particles, transit through the sample cell but only transect the laser at a few angles.

L311: Which aerosol concentration do you refer to? Volume, number or mass?

It is more accurate to say this phenomenon would occur due to a rapidly changing $\beta_{scat}$, and that the e.g. rapidly increasing $\beta_{scat}$ would occur due to an increasing aerosol number (or mass or volume) concentration. We have modified the text as follows:

Another potential source of error during ambient measurements is a rapidly changing $\beta_{scat}$, e.g. when there is an increase in the aerosol number concentration. If the sampled $\beta_{scat}$ increases rapidly, the $\beta_{scat}$ gradient within the sample cell will be observed in the scattering phase function. For example, if the $\beta_{scat}$ at the instrument inlet increases during a measurement, there may be more scattering in the section of the sample cell corresponding to forward scattering angles than in the portion of the cell corresponding to backscattering.

L313: "in" should be "if".

This has been fixed, thank you.

L341: How did other imaging nephelometer perform in the comparison with AOP-derived scattering?

We have included the following:

If we take the AOP-derived scattering as a truth measurement, i.e. without its own error, we can say that the LiNeph is precise within <2%, although with a positive bias of ~30%, likely due to calibration error. This is consistent with the reported precision of similar techniques, i.e. the PI-Neph reports agreement with commercial integrating nephelometers to within 5% (Espinosa et al., 2017).

L363: It's not "cumulative".

We have corrected the text as follows:

Supplementary Fig. S11 shows the average normalized number-weighted size distributions for both transects as measured by the LAS with an applied ammonium sulfate calibration (Moore et al., 2021).

L372: It would be better to present the measured values of the asymmetry parameter in the main text besides in the figure.

We have modified the text as follows:

The asymmetry parameter, described below, increases from 0.568 to 0.620, showing an increase in forward scattering that is consistent with increasing particle size. However, it is important to note that changes in particle composition, hence refractive index, have also been observed as a consequence of photochemical aging in biomass burning aerosol.

L416: It's unclear that "accurately determine the asymmetry parameter within 3%".

We have modified the text as follows:

Line 455:

The phase function measurements (e.g. Fig. 7) allow for precise measurements of the asymmetry parameter with a relative standard deviation of less than 3%. But, as discussed in Section 2.4, the geometry of the LiNeph requires that the lasers be offset from the optical axis of the wide angle lenses, introducing a non-zero η. Based on measurements of PSLs (Fig. S2-S5), we can set an upper bound of this effect on the measured σ°. We used the same method to investigate the effect of η on the measured asymmetry parameter. Supplementary Figure S12 shows that for polydisperse lognormal

aerosol size distributions centered around 200 nm, with varying refractive indices, the effect of a non-zero η on g is a bias of less than 2%. For the largest modeled aerosol population, with a mode centered at 400 nm, the bias was 5%. This effect is small in part because the geometry of the instrument results in offsetting biases when $\sigma°_{\text{"Perp"}}$ and $\sigma°_{\text{"Para"}}$ are combined to calculate $P_{11}$. For the four $P_{11}$ measurements of PSL calculated from the data in Supplemental Fig. S2-S5, the average ratio of measured to Mie-calculated g was $1.01 \pm 0.09$.

Line 482:

Finally, we showed that we can precisely (less than 3% relative standard deviation) and accurately (within 10% for the PSLs examined in this work) determine the asymmetry parameter.

---

## Author Comment (AC3)

[Figure]

**Figure 4. Phase function (P₁₁, top) and degree of linear polarization (-P₁₂/P₁₁, bottom) for λ = 660 nm light scattered by D_p**
= 900 nm PSLs. Measurements are shown as red circles and the values calculated from Mie theory are shown as solid lines.
The gold line shows the values calculated if scattering plane rotation (η) is assumed to be zero. If we use an upper estimate
of η, Mie theory predicts that the observations would follow the teal line. Stray light from the inside of the instrument
introduces noise at scattering angles around 30° and 135° for both data products.

---

## Author Comment (AC4)

**Laser Imaging Nephelometer for aircraft deployment for FIREX-AQ**

Adam T. Ahern[1,2], Frank Erdesz[1,2], Nicholas L. Wagner[1,2]*, Charles A. Brock[1], Ming Lyu[3], Kyra Slovacek[2,4], Richard H. Moore[5], Elizabeth B. Wiggins[5,6], and Daniel M. Murphy[1]

[1]NOAA Chemical Sciences Laboratory, Boulder, CO 80305, USA
[2]Cooperative Institute for Research in Environmental Sciences, University of Colorado, Boulder, CO 80309, USA
[3]Department of Chemistry, University of Alberta, Edmonton, AB T6G 2B4, Canada
[4]Civil, Environmental, and Architectural Engineering, University of Colorado, Boulder, CO 80309, USA
[5]NASA Langley Research Center, Hampton, VA 23666, USA
[6]NASA Postdoctoral Program, Universities Space Research Association, Columbia, MD 21046, USA
*Now at Ball Aerospace, Westminster, CO 80021, USA

*Correspondence to*: Adam T. Ahern (adam.ahern@noaa.gov)

**Supplemental Information**

Figure S1 shows a schematic of the laboratory setup for calibrating the Laser imaging nephelometer (LiNeph). In this configuration, the sample volume of the LiNeph can be evacuated and back-filled with calibration gases, either He or $CO_2$. Alternatively, calibration aerosol may be introduced via positive pressure from a compressed zero air cylinder by removing the pump from the system and allowing the LiNeph to exhaust through the filter into the room.

Figure S2, S3, S4, and S5 show a comparison of measured differential scattering coefficients ($\sigma°$) and calculations from Mie theory. In the bottom panels, two Mie theory calculations of $I_{scat}$ are shown to evaluate the magnitude of the scattering plane rotation shown in Fig. 1b as $\eta$. The calculations are based on Equations 81 and 82 from Dolgos (2014),

$$I_{scat,j}(\theta) \propto \beta_{scat} * (P_{11}(\theta) + (P_{12}(\theta) * q_j^{in}(\theta))) \tag{S1},$$

where $I_{scat,j}$ is the radiance observed by the detector j (i.e. "Perp" or "Para"), $\beta_{scat}$ is the integrated scattering coefficient, $P_{11}$ and $P_{12}$ are scattering matrix elements, and $q_j^{in}$ is the second Stokes parameter for the incident light in the scattering plane. When the central axis of the wide angle lens is aligned with the polarization of the laser, $q_j^{in}$ equals $q_j$. However, because of the small offset that allows for two wavelengths to be measured simultaneously, the rotation of the scattering plane must be accounted for using,

$$q_j^{in}(\theta) = q_j * cos(2\eta^{\theta,j}) + u_j * sin(2\eta^{\theta,j}) \tag{S2},$$

where $q_j$ and $u_j$ are the Stokes parameters for the light as defined by the lasers' polarization plane. $\eta$ is the angle, in the scattering plane that is created by the offset of the lasers from the wide angle lens' central axis. In the first calculation of $I_{scat,j}(\theta)$, shown as a gold line in S2-S5, we assume that $\eta$ is zero, which results in $I_{scat,j}$ as shown in Eq. 4 and 6.

For the second calculation, we use an estimated $\eta$, shown in the top-most panels of S2 and S3. We estimated $\eta$ by first locating the origin pixel on the CCD array for each camera using an imaged grid. Once we knew the origin pixel row location, we applied the same angular scattering calibration that we found using PSLs to the pixel row (instead of the pixel column). We

believe this to be a reasonable estimate given the radially symmetric nature of the wide angle lens, the square shape of the CCD pixels, and the mounting orientation of the CCD. We can then estimate η using the peak location of the Gaussian fit for each CCD orientation, j. The top-most panels of S2 and S3 show that this estimate of η is a function of the detector geometry, j, and scattering angle, θ. For this work, we optimized the laser alignment, which changes the laser position and therefore η, to minimize background light in the instrument. In future work, it would be prudent to minimize η to the extent practical, or else ensure that η is the same for both "Para" and "Perp" detectors.

Figure S2 S6 shows the scattering angle calibration performed by matching maxima and minima observed in measured scattering intensity to the maxima and minima predicted by Mie theory. The calibration aerosol was were atomized and dried polystyrene latex spheres of known sizes.

Figure S3 S7 and Fig. S4 shows a series of measurements from of pure $CO_2$. For this experiment, the LiNeph chamber was filled with $CO_2$ and then a series of images were captured. From these images, the He image was subtracted. Gaussian fits were applied to each pixel column. Figure S3 S7 shows the area under that Gaussian fit for each image as a function of time for an individual pixel column, which corresponds to a scattering angle.

Figure S4 S8 shows the standard deviation for the area under fitted Gaussian curves as a function of the average area under Gaussian curves for $CO_2$ measurements. The two traces show different exposure durations and have been normalized by CCD exposure time to be proportional to scattering intensity instead of the digital output. The spread in the data points is from additional noise due to stray light within the instrument. This illustrates that for measurements with abundant signal, the standard deviation is typically <2% of the average signal. But in cases of low signal, the standard deviation becomes constant ~200 bits$^2$, and uncertainty is likely dominated by electronic noise. Since phase functions can span many orders of magnitude, we find it advantageous at times to increase the exposure time of the CCD, which linearly increases the signal while the electronic noise remains the same. The resulting Gaussian fit values can be scaled by their exposure time and recombined, allowing for more precise measurements of phase functions spanning multiple orders of magnitude.

Figure S5 S9 shows the sum of all pixels while sampling a well-mixed smoke plume during FIREX-AQ. The red colored region indicates the period of the sampling when a filter is in the sampling line. Only the steady-state data points of the red region are used to correct for changing gas-phase scattering and instrument background light.

Figure S6 S10 shows a subset of the plot on the right side of Fig. 6, showing only the data where the bridge dilution system was not being used, see Fig. 3. The greater linearity between the integrated scattering measured by the LiNeph and the AOP instrument suite for undiluted data compared to all data is consistent with the dilution system being an added source of uncertainty. This plot also indicates that there may be an error in the magnitude of the differential scattering calibration which results in the LiNeph measuring 20-30% more light scattering than what is observed by the AOP instrument suite. We do not expect this to influence the polarimetry measurements as they are normalized functions.

Figure S7 S11 markers show the average normalized number-weighted size distributions for two transects during the Williams Flats Fire on August 7$^{th}$, 2019. Each size distribution was first normalized by its maximum peak height. Then the average and

standard deviation of each size bin was calculated for all the normalized size distributions from each transect. Size distributions were measured using the LAS with an ammonium sulfate calibration as discussed in Moore et al. (2021). Additionally, we fit

70   each individual size distribution with a lognormal distribution and then calculated the average and standard deviation of the fitted mode. Transect 1 had an average mode of $0.174 \pm 0.014\ \mu m$ and Transect 10 had an average mode of $0.225 \pm 0.011\ \mu m$. Figure S12 shows the asymmetry parameter (g) as calculated by MiePlot for various theoretical aerosol populations (Laven, 2003). We varied the complex refractive index and the mode of the lognormal distribution to simulate what effect a non-zero $\eta$ will have on the asymmetry parameter measured by the LiNeph for different aerosol populations. The x-axis shows the g for

75   $\eta = 0$. The y-axis shows the g that is calculated when the upper limit of $\eta$ used to calculate to the $\sigma^\circ$ observed by each detector. We then use the $\sigma^\circ$ to calculate $P_{11}$ (incorrectly assuming $\eta = 0$) to evaluate the error that might be introduced by neglecting to account for $\eta$ when calculating g. We see that for a mode diameter of 200 nm, changing the complex refractive index induces small changes in g, and that the potential bias from $\eta$ is about 2% for $\lambda = 660$ and 405 nm, albeit with different signs. As the mode diameter changes, the sign of the bias can change, but for the cases we examined the bias does not exceed a magnitude

80   of 5%.

Dolgos, G.,: Polarized Imaging Nephelometer Development and Applications on Aircraft, Doctor of Philosophy, Physics, University Maryland Baltimore County, 2014.

Laven, P.: Simulation of rainbows, coronas, and glories by use of Mie theory, Appl. Optics, 42, 436–

85   444, https://doi.org/10.1364/AO.42.000436, 2003.

Laven, P.: MiePlot, available at: http://www.philiplaven.com/mieplot.htm, last access: 14 Oct 2021.

[Figure]

**Figure S1. Diagram of laboratory calibration of Laser Imaging Nephelometer.**

[Figure]

Figure S2. Comparison of measured versus calculated differential scattering coefficient ($\sigma°$) of 900 nm PSLs for $\lambda = 405$. a) Dashed (solid) lines show the estimated scattering plane rotation ($\eta$) based on pixel row for 405 nm laser as viewed by the "Perp" ("Para") oriented CCD. b) Markers show the measured $\sigma°$, the solid lines show the $\sigma°$ calculated using Mie theory. The gold lines were calculated assuming $\eta = 0$ and the green were calculated using an estimate of $\eta$.

[Figure]

**Figure S3. Same as 2 but for λ = 660 nm and PSLs with D_p = 900 nm.**

[Figure]

**Figure S4. Same as S2 but for λ = 405 nm and PSLs with $D_p$ = 600 nm.**

[Figure]

**Figure S5. Same as S2 but for λ = 660 nm and PSLs with $D_p$ = 600 nm.**

[Figure]

**Figure S6.** Calibration for converting pixel columns to scattering angle using local maxima and minima in σ° from PSLs as measured by a) "Perp" and b) "Para" cameras. Markers show maxima and minima from Mie theory as a function of the corresponding pixel column of the measured σ°, with the differential scattering coefficient calibration applied. Error bars indicate the 95% confidence interval from the linear fit. Dots show the pixel column where the maxima/minima would be located in the raw data, without the differential scattering coefficient calibration.

[Figure]

**Figure S7.** Laser stability shown as area under Gaussian fit for light scattered by $CO_2$ at two scattering angles. Pink circles and yellow squares show individual measurements of light scattering at 10° and 104°, respectively, as a function of time.

[Figure]

115 **Figure S4S8. Standard deviation of the area under Gaussian fit as a function of the average area under the Gaussian fit for two exposure durations. Each symbol represents a scattering angle for a series of measurements. Black lines are linear regressions fits.**

[Figure]

120 **Figure S5S9. Change in light scattering after inserting a filter during smoke sampling. Red asterisks indicate the filter period and purple squares indicate sample data. Note: only steady-state data from the filter period is used for the subtraction of gas-phase scattering from the sample data.**

[Figure]

**Figure S10. Integrated scattering for undiluted measurements only.**

[Figure]

130

**Figure S11.** Average normalized particle size distribution for two transects of the Williams Flats fire. Shaded area indicates one standard deviation from the average. The average ± standard deviation geometric mean diameter ($D_g$) is calculated from lognormal fits of each measured size distribution during each transect.

[Figure]

135 **Figure S12. Asymmetry parameter (g) calculated for theoretical aerosol populations. The populations vary in the assumed complex refractive indices and the mode of the lognormal polydisperse size distribution. The x-axis shows the g expected to be measured if $\eta$ = 0. The y-axis shows the g that would be measured by the LiNeph, assuming the estimated $\eta$ were used to calculated the differential scattering coefficients ($\sigma°$), but then $\eta$ = 0 were used to calculate g.**